# Insights from aquaporin structures into drug-resistant sleeping sickness

**Modestas Matusevicius**[1†], **Robin A Corey**[2,3†], **Marcos Gragera**[4†], **Keitaro Yamashita**[5†], **Teresa Sprenger**[6†‡], **Marzuq Ungogo**[7,8§], **James N Blaza**[9], **Pablo Castro-Hartmann**[10], **Dimitri Y Chirgadze**[6], **Sundeep Chaitanya Vedithi**[1], **Pavel Afanasyev**[11], **Roberto Melero**[4], **Rangana Warshamanage**[5#], **Anastasiia Gusach**[5], **José-Maria Carazo**[4], **Mark Carrington**[6], **Tom Blundell**[1], **Garib N Murshudov**[5], **Phillip J Stansfeld**[12], **Mark SP Sansom**[2], **Harry P De Koning**[7], **Christopher G Tate**[5], **Simone Weyand**[1,6,13,14]*

[1]Department of Medicine, University of Cambridge, Victor Phillip Dahdaleh Building, Heart and Lung Research Institute, Cambridge, United Kingdom; [2]Department of Biochemistry, University of Oxford, Oxford, United Kingdom; [3]School of Physiology, Pharmacology and Neuroscience, Biomedical Sciences Building, Bristol, United Kingdom; [4]Instruct Image Processing Centre (I2PC), Unidad de Biocomputación, Centro Nacional de Biotecnología, Madrid, Spain; [5]MRC Laboratory of Molecular Biology, Cambridge, United Kingdom; [6]Department of Biochemistry, University of Cambridge, Cambridge, United Kingdom; [7]School of Infection and Immunity, University of Glasgow, Glasgow, United Kingdom; [8]Department of Veterinary Pharmacology and Toxicology, Ahmadu Bello University, Zaria, Nigeria; [9]Structural Biology Laboratory and York Biomedical Research Institute, Department of Chemistry, The University of York, York, United Kingdom; [10]Materials and Structure Analysis, Thermofisher Scientific, Eindhoven, Netherlands; [11]ETH Zurich, ScopeM, Cryo-EM Knowledge Hub, Zurich, Switzerland; [12]School of Life Sciences, Gibbet Hill Campus, The University of Warwick, Coventry, United Kingdom; [13]Cambridge Institute for Medical Research, Keith Peters Building, Biomedical Campus, Cambridge, United Kingdom; [14]EMBL's European Bioinformatics Institute (EMBL-EBI), Wellcome Genome Campus, Cambridgeshire, United Kingdom

*For correspondence:
sw644@cam.ac.uk

[†]These authors contributed equally to this work

**Present address:** [‡]Eurofins BioPharma Product Testing Germany, Munich, Germany; [§]Division of Immunology, The Roslin Institute, University of Edinburgh, Edinburgh, United Kingdom; [#]Science & Technology Facilities Council, Research Complex at Harwell, Didcot, United Kingdom

**Competing interest:** The authors declare that no competing interests exist.

## eLife Assessment

In this **important** study, the authors set out to determine the molecular interactions between the AQP2 from *Trypanosoma brucei* (TbAQP2) and the trypanocidal drugs pentamidine and melarsoprol to understand how TbAQP2 mutations lead to drug resistance. Using cryo-EM, molecular dynamics simulations, and lysis assays the authors present **convincing** evidence that mutations in TbAQP2 make permeation of trypanocidal drugs energetically less favourable, and that this impacts the ability of drugs to achieve a therapeutic dose. Overall, this data will be of interest for those working on aquaporins, and development of trypanosomiasis drugs as well as drugs targeting aquaporins in general.

**Abstract** *Trypanosoma brucei* is the causal agent of African trypanosomiasis in humans and animals, the latter resulting in significant negative economic impacts in afflicted areas of the world. Resistance has arisen to the trypanocidal drugs pentamidine and melarsoprol through mutations in the aquaglyceroporin TbAQP2 that prevent their uptake. Here, we use cryogenic

electron microscopy to determine the structure of TbAQP2 from *T. brucei*, bound to either the substrate glycerol or to the sleeping sickness drugs, pentamidine or melarsoprol. The drugs bind within the AQP2 channel at a site completely overlapping that of glycerol. Mutations leading to a drug-resistant phenotype were found in the channel lining. Molecular dynamics (MD) simulations showed the channel can be traversed by pentamidine, with a low energy binding site at the centre of the channel, flanked by regions of high energy association at the extracellular and intracellular ends. Drug-resistant TbAQP2 mutants are still predicted to bind pentamidine, but the much weaker binding in the centre of the channel observed in the MD simulations would be insufficient to compensate for the high energy processes of ingress and egress, hence impairing transport at pharmacologically relevant concentrations. The structures of drug-bound TbAQP2 represent a novel paradigm for drug–transporter interactions and are a new mechanism for targeting drugs in pathogens and human cells.

## Introduction

Human African trypanosomiasis (HAT) remains a neglected tropical disease (***World Health Organisation, 2023***). It is transmitted by tsetse flies and, once symptoms are apparent, treatment is necessary to avoid fatality. The infection starts with a haemolymphatic stage that progresses to a late stage infection of the central nervous system, at which point severe neurological symptoms occur with rapid progression to coma and death (***Kennedy, 2013***). Early-stage infections are commonly treated with pentamidine and the late stage with melarsoprol, although for most cases this is now being replaced with fexinidazole (***De Koning, 2020***). Animal African trypanosomiasis (AAT) causes billions of dollars in damage to agriculture throughout the tropics (***Giordani et al., 2016***; ***Desquesnes et al., 2013***). For both HAT and AAT, chemotherapy remains the only control option and most of the medication is old, inadequate and threatened by resistance (***De Koning, 2020***; ***Giordani et al., 2016***; ***Baker et al., 2013***; ***Carruthers et al., 2021***).

For most trypanocidal drugs, resistance is associated with mutations in the membrane transporters through which the drugs enter the trypanosome (***Munday et al., 2015b***). The uptake of pentamidine and melarsoprol was initially attributed to a High Affinity Pentamidine Transporter (HAPT) (***Bridges et al., 2007***; ***De Koning, 2001***), which was subsequently identified as the aquaglyceroporin TbAQP2 (***Baker et al., 2012***; ***Munday et al., 2014***). Deletions and chimeric rearrangements between TbAQP2 and the adjacent gene TbAQP3 were found to be responsible for melarsoprol–pentamidine cross-resistance (MPXR; ***Figure 1***; ***Baker et al., 2013***; ***Munday et al., 2014***; ***Graf et al., 2015***; ***Graf et al., 2013***; ***Pyana Pati et al., 2014***). To understand the differences between TbAQP2 (water/glycerol/drug transporter) and TbAQP3 (classical aquaglyceroporin transporting water/glycerol ***Uzcategui et al., 2004***) but not pentamidine or melarsoprol (***Baker et al., 2012***; ***Munday et al., 2014***; ***Alghamdi et al., 2020***), mutants were made exchanging amino acids between the proteins (***Alghamdi et al., 2020***). Mutation of TbAQP2 to add any of the TbAQP3 selectivity filter residues Tyr250, Trp102, or Arg256 resulted in inhibition of melarsoprol and pentamidine uptake, whereas TbAQP3 containing all three of their TbAQP2 counterparts (Leu250, Ile102, and Leu256) permitted the uptake of pentamidine. However, the molecular details of the interactions between TbAQP2 and drugs as well as the resistance mechanism remained unclear, with alternative models proposing that TbAQP2 acts as a channel (***Munday et al., 2015b***; ***Alghamdi et al., 2020***; ***Munday et al., 2015a***) or receptor (***Song et al., 2016***) for the drugs. To resolve this quandary and to resolve the molecular interactions underlying drug uptake by aquaporins, we determined the structures of TbAQP2 bound to either melarsoprol, pentamidine, or the native substrate, glycerol.

## Results
### Cryo-EM structures of TbAQP2 and drug binding

TbAQP2 was expressed using the baculovirus expression system in insect cells and purified in the presence of either a drug (melarsoprol or pentamidine) or the substrate glycerol (see Methods, *Figure 2—figure supplement 1a, b*). Structures of TbAQP2 bound to either glycerol, melarsoprol, or pentamidine were determined by electron cryogenic microscopy (cryo-EM) to overall resolutions of 3.2, 3.2, and 3.7 Å, respectively (*Table 1*, *Figure 2—figure supplement 1c*, *Figure 2—figure*

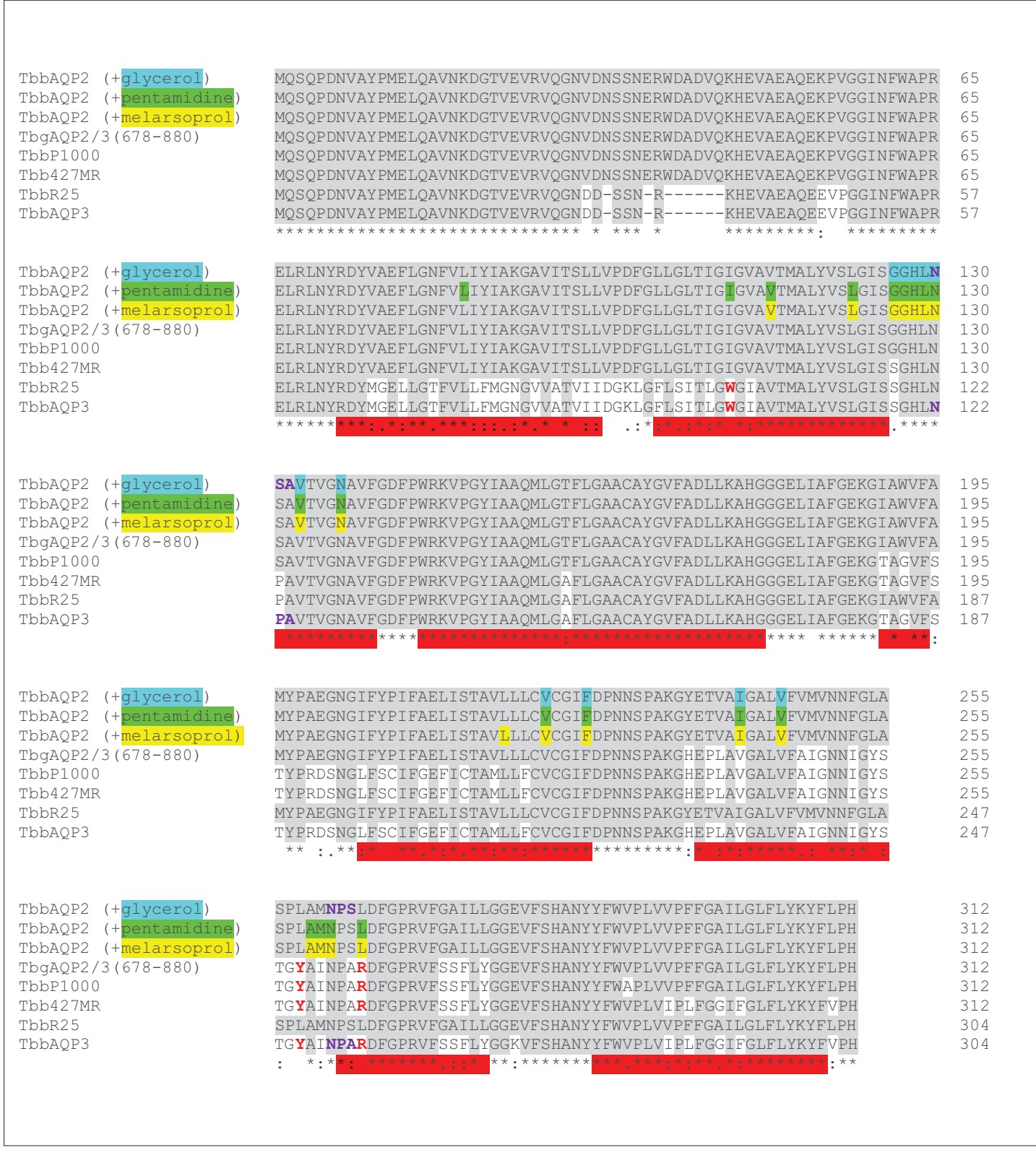

**Figure 1.** Alignment of aquaporin amino acid sequences. Amino acid sequences of AQP2 and AQP3 from *Trypanosoma brucei brucei* (Tbb) are aligned with chimeric aquaporins from drug-resistant strains (P1000, 427MR, and R25) and a chimera from drug-resistant *T. b. gambiense* (TbgAQP2/3). Three identical sequences are shown of TbbAQP2 with residues making contact to either glycerol, pentamidine, or melarsoprol highlighted (cyan, green, or yellow, respectively). Red bars represent transmembrane regions. Residues highlighted in grey are conserved in all sequences. The conserved NSA/NPS motif in TbAQP2 and the analogous NPA/NPA motif in TbAQP3 are in purple font and the residues mutated for the molecular dynamics (MD) simulation

*Figure 1 continued on next page*

*Figure 1 continued*

studies (I110W, L258Y, and L264R) are in red. Sequence information was obtained from *Munday et al., 2014* except R25 (*Unciti-Broceta et al., 2015*) and TbgAQP2/3 (678–800) (*Pyana Pati et al., 2014*).

**Table 1.** Cryo-EM data collection, refinement, and validation statistics.

| Data collection and processing | TbAQP2 + glycerol (EMDB: EMD-16864) (PDB: 8OFZ) | TbAQP2 + melarsoprol (EMDB: EMD-16862) (PDB: 8OFX) | TbAQP2 + pentamidine (EMDB: EMD-16863) (PDB: 8OFY) |
|---|---|---|---|
| Magnification | 130,000 | 130,000 | 130,000 |
| Voltage (kV) | 300 | 300 | 300 |
| Electron exposure ($e^-/Å^2$) | 25.2 | 57.6 | 57.6 |
| Defocus range (μm) | [−0.6, −4] | [−0.6, −3.6] | [−0.8, −4] |
| Pixel size (Å) | 1.07 | 1.042* | 1.042* |
| Symmetry imposed | C4 | C4 | C4 |
| Initial particle images (no.) | 849,607 | 1,927,102 | 1,263,534 |
| Final particle images (no.) | 849,607 | 126,551 | 83,845 |
| Map resolution (Å) | 3.2 | 3.2 | 3.7 |
| FSC threshold | 0.143 | 0.143 | 0.143 |
| Map resolution range (Å)[†] | 3.1–4.4 | 3.0–8.3 | 3.5–9.8 |
| **Refinement** | | | |
| Initial model used | AlphaFold | AlphaFold | AlphaFold |
| Model composition in the ASU | | | |
| Non-hydrogen atoms | 1842 | 1852 | 1855 |
| Protein residues | 244 | 244 | 244 |
| Substrates | 2 | 1 | 1 |
| *B* factors ($Å^2$) | | | |
| Protein | 159.7 | 140.2 | 170.4 |
| Ligand | 180.8 | 175.0 | 209.6 |
| R.m.s. deviations | | | |
| Bond lengths (Å) | 0.0074 | 0.0099 | 0.0060 |
| Bond angles (°) | 1.730 | 1.903 | 1.525 |
| Validation | | | |
| MolProbity score | 1.99 | 1.79 | 1.79 |
| Clashscore | 4.34 | 6.80 | 4.75 |
| Poor rotamers (%) | 5.52 | 1.66 | 2.21 |
| Ramachandran plot | | | |
| Favoured (%) | 96.69 | 96.28 | 95.87 |
| Allowed (%) | 3.31 | 3.72 | 3.72 |
| Disallowed (%) | 0 | 0 | 0.41 |

*Original pixel size in movies: 1.07 Å/px. Calibrated for modelling.
[†]Local resolution range.

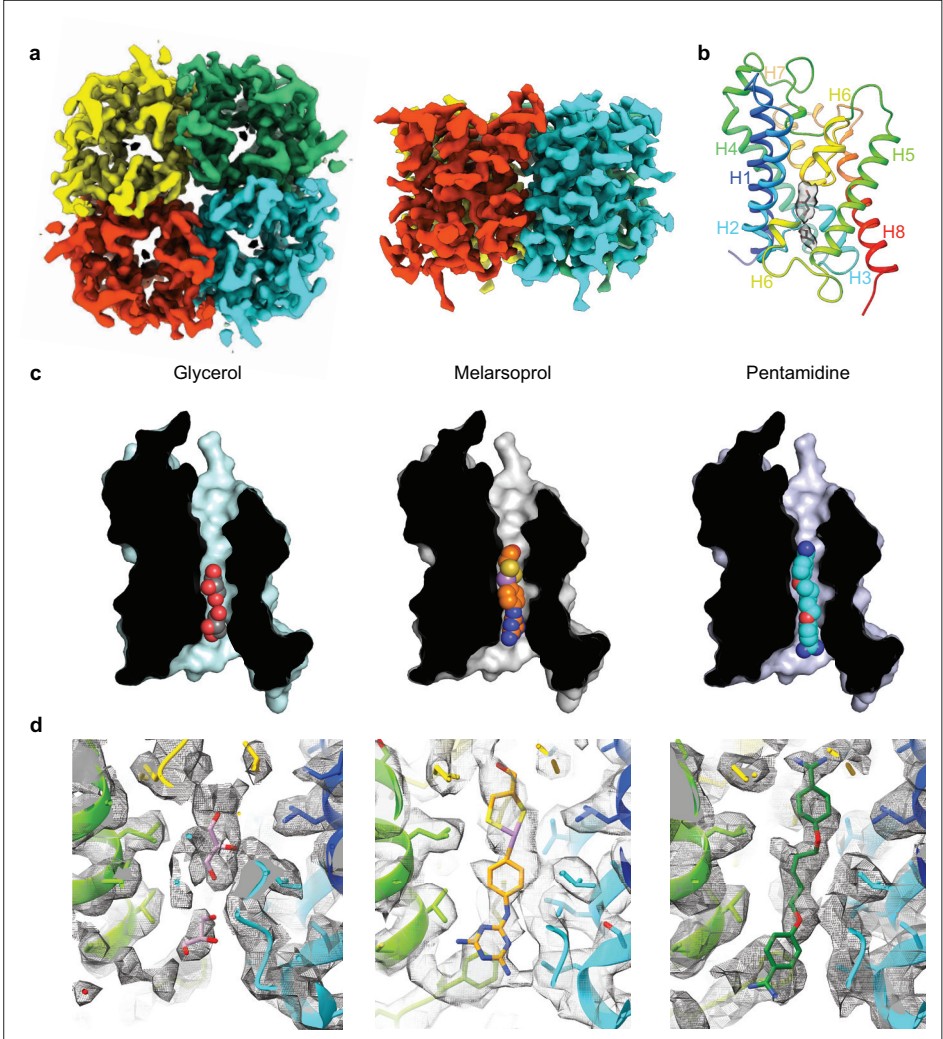

**Figure 2.** Cryo-EM structures of TbAQP2 bound to either glycerol, melarsoprol, or pentamidine. (**a**) Overall structure of the TbAQP2 tetramer viewed either from the extracellular surface or within the membrane plane. (**b**) Structure of protomer A of TbAQP2 viewed as a cartoon (rainbow colouration) with glycerol (sticks) bound. The cryo-EM density for glycerol is shown as a grey surface. (**c**) Cut-away views of the channel in each of the TbAQP2 structures showing bound substrates and drugs (spheres) with atoms coloured according to type: red, oxygen; yellow, sulphur; blue, nitrogen; purple, arsenic; carbon, grey (glycerol), orange (melarsoprol), or cyan (pentamidine). (**d**) Cryo-EM densities (grey surface) for glycerol, melarsoprol, and pentamidine in their respective structures. See *Figure 2—figure supplements 6 and 7* for different views of the substrates and comparisons between densities.

The online version of this article includes the following figure supplement(s) for figure 2:

**Figure supplement 1.** Purification of TbAQP2 and identification of the drug binding site by hotspot analysis.

**Figure supplement 2.** Flow chart of the cryo-EM image processing and structure determination.

**Figure supplement 3.** Local resolution maps of TbAQP2 structures.

**Figure supplement 4.** Density maps of secondary structure elements in the TbAQP2 structures.

**Figure supplement 5.** Density half maps of ligands bound in the TbAQP2 structures.

**Figure supplement 6.** Depictions of the side chain and substrate densities.

**Figure supplement 7.** Comparisons between substrate densities.

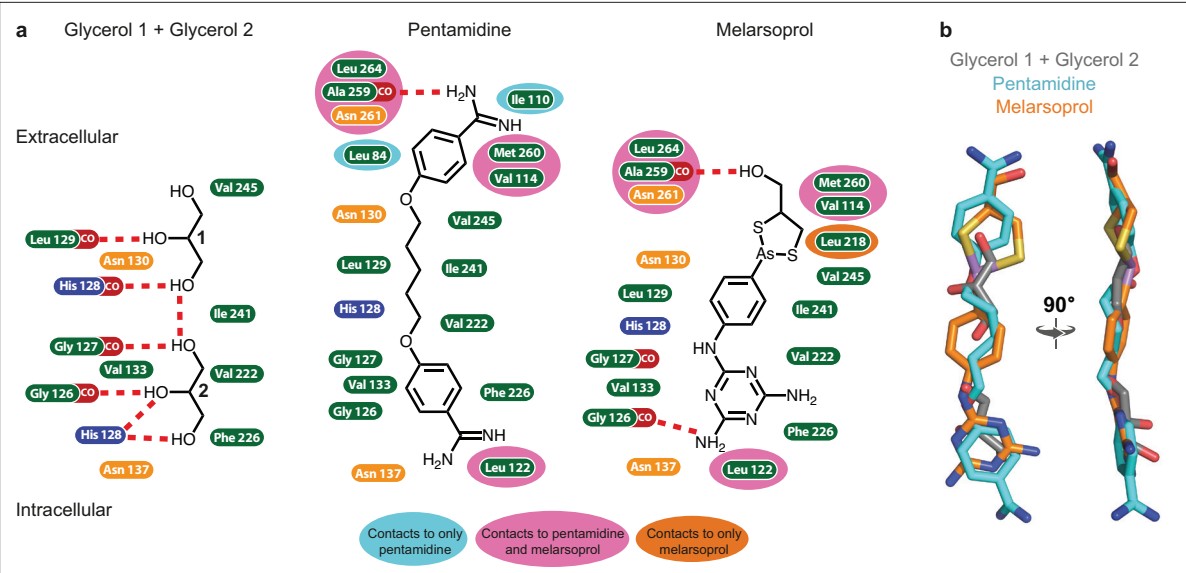

**Figure 3.** Interactions between TbAQP2 and bound substrates and drugs. (**a**) Amino acid residues containing atoms ≤3.9 Å from substrate or drug model in the cryo-EM structures are depicted and coloured according to the chemistry of the side chain: hydrophobic, green, positively charged, blue; polar, orange. Interactions with a backbone carbonyl group are shown as CO (red) and potential hydrogen bonds are shown as red dashed lines. Residues highlighted in an additional colour (grey, pale blue, pink, or dark orange) make contacts in two or less of the structures (as indicated on the figure), whilst those without additional highlighting make contacts in all three structures. (**b**) Protomer A from each structure was aligned and the positions of the two glycerol molecules (grey), melarsoprol (orange), and pentamidine (cyan) are depicted.

supplements 2 and 3a-c). The resolution was sufficient to see the majority of the side chains and the bound drug or glycerol (*Figure 2—figure supplements 4–7*). The cryo-EM structures of TbAQP2 (excluding the drugs/substrates) were virtually identical (rmsd 0.2 Å) and represented the aquaporin tetramer (*Figure 2a, b*) that is typical for this family of channels (*Gonen and Walz, 2006*). The tetramer contained four channels, each consisting of a single TbAQP2 polypeptide folded into a bundle of eight α-helices (*Figure 2b*), two of which (helices H3 and H7) penetrated the plasma membrane only partially, as is typical for aquaporins (*Gonen and Walz, 2006*), the others crossing the membrane fully.

Melarsoprol and pentamidine are bound in a channel formed by the TbAQP2 protomer, completely overlapping the binding site for the substrate glycerol (*Figure 2c, d*). The densities of the substrates were distinct from one another and are consistent with the structures of the substrates (*Figure 2—figure supplements 6 and 7*). The position of drug binding is consistent with a hotspot analysis performed on the structures (see Methods and *Figure 2—figure supplement 1d, e*). This analysis defines likely regions where molecules may bind based on in silico docking scores of small molecule fragments docking to a model of TbAQP2 (*Radoux et al., 2016*) (see Methods). Binding of the two glycerol molecules is mediated by a total of six hydrogen bonds, four to backbone carbonyl groups (residues Gly126, Gly127, His128, and Leu129) and two to the side chain of His128 (*Figure 3a*). These are on a polar strip of residues running down the channel, that also includes Val133, which makes van der Waals contacts with the glycerol molecule closest to the cytoplasm. The remainder of the contacts are van der Waals interactions (Val222, Phe226, Ile241, and Val245) on the opposite side of the channel. It is striking that all eleven amino acid residues within 3.9 Å of the two glycerol molecules also make contacts to both pentamidine and melarsoprol (*Figure 3a*). Whereas the TbAQP2-glycerol contacts are equally polar and non-polar, van der Waals interactions dominate contacts between the drugs and TbAQP2. The exceptions are a hydrogen bond between the backbone carbonyl of Ala259 and both drugs, and a backbone carbonyl hydrogen bond between Gly126 and melarsoprol that is also seen between glycerol and TbAQP2 (*Figure 3a*). It is also remarkable that an overlay between the three different structures shows a very close alignment between glycerol, pentamidine and melarsoprol (*Figure 3b*) which is consistent with the channel structure being virtually identical regardless of whether a drug or glycerol is bound. The size of the channel with pentamidine bound is very similar to the glycerol-bound *Plasmodium falciparum* AQP structure (*Figure 2—figure supplement 1f*). The close fit of the drugs within the channel is consistent with the extremely limited modification of the

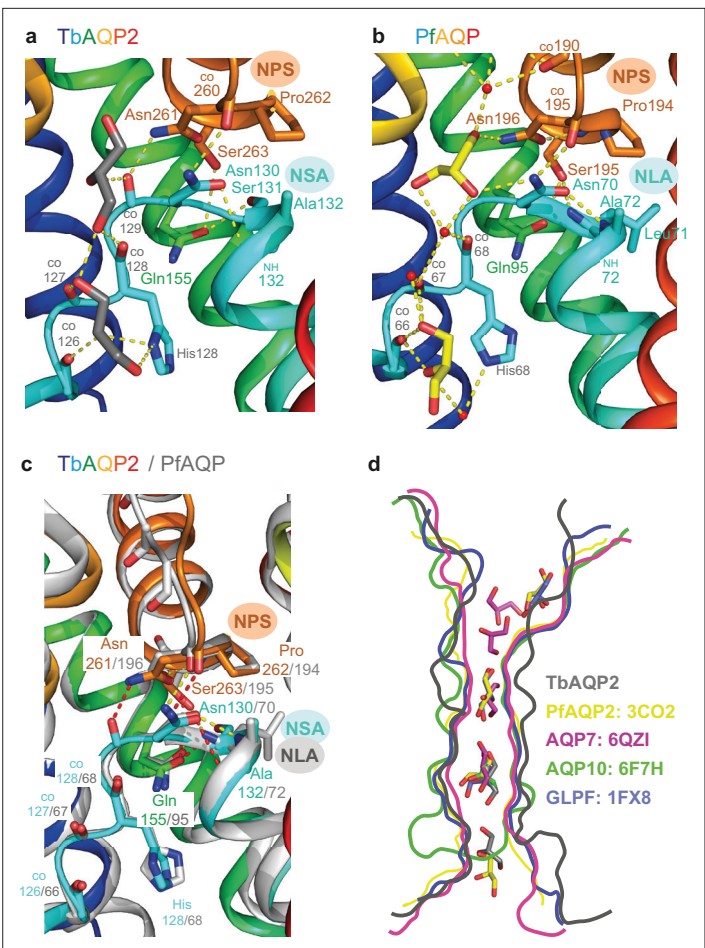

**Figure 4.** Comparison between conserved glycerol-binding regions of TbAQP2 and PfAQP. (**a**) The structure of TbAQP2 is depicted (rainbow colouration) with residues shown that make hydrogen bonds (yellow dashed lines) to either glycerol (grey) or in the NPS/NSA motifs. CO, backbone carbonyl groups involved in hydrogen bond formation. (**b**) The structure of PfAQP2 is depicted (rainbow colouration) with residues shown that make hydrogen bonds (yellow dashed lines) to either glycerol (yellow sticks) or water (red spheres), or in the NPS/NLA motifs. CO, backbone carbonyl groups involved in hydrogen bond formation. (**c**) Structures of TbAQP2 (rainbow colouration) and PfAQP are superimposed to highlight structural conservation of the NPS and NSA/NLA motifs. (**d**) Outline of channel cross-sections from aquaglyceroporins containing bound glycerol molecules (sticks): TbAQP2 (grey); PfAQP (yellow; PDB ID 3CO2); human AQP7 (magenta, PDB ID 6QZI); human AQP10 (green, PDB ID 6F7H); and *Escherichia coli* GlpF (purple, PDB ID 1FX8).

drugs that is possible whilst retaining the ability to pass through the pore, including modifications that reduce flexibility (*Alghamdi et al., 2020*). Diamidines with fixed-angle structures such as diminazene and furamidine are not substrates for TbAQP2 but are exclusively accumulated through aminopurine transporter TbAT1 (*de Koning et al., 2004*; *Ward et al., 2011*). A number of amino acid residues interact only with pentamidine and/or melarsoprol and not with glycerol (Leu84, Ile110, Val114, Leu122, Leu218, Ala259, Asn261, Met260, and Leu264), which is a result of the drugs being longer and extending further along the channel towards the extracellular surface when compared with glycerol (*Figures 2 and 3*). Towards the intracellular surface, only Leu122 makes contacts to the drugs and not to glycerol. In all, 19 residues in TbAQP2 interact with pentamidine, compared with 18 residues interacting with melarsoprol and 11 to the two molecules of glycerol.

The structure of AQP2 from *Plasmodium falciparum* (PfAQP) (*Newby et al., 2008*) with glycerol bound is the closest structural homologue to TbAQP2 available (rmsd 0.8 Å, PDB ID 3CO2, 33% amino acid sequence identity). The overall arrangement of α-helices in TbAQP2 is identical to that in PfAQP (*Figure 4a–d*) and other aquaglyceroporins whose structures have been determined (human AQP7 *de Maré et al., 2020*), human AQP10 (*Gotfryd et al., 2018*), and *Escherichia coli*

GlpF (*Fu et al., 2000*). Despite the low identity in amino acid sequence, there are striking similarities between the structures of TbAQP2 and PfAQP in the conserved NPA/NPA motif and the positions of bound glycerol molecules in the channel. The NPA/NPA motif is found at the ends of the half-helices H3 and H7, with TbAQP2 and PfAQP similarly differing from the canonical sequences (NPS/NSA in TbAQP2 and NPS/NLA in PfAQP; *Figure 4a, b*). Superposition of TbAQP2 and PfAQP shows the position and rotamer of Asn–Pro–Ser in the NPS motif and both Asn and Ala in the NSA/NLA motifs are identical (*Figure 4c*). Some of the network of hydrogen bonds that maintain the structure of this region are also conserved, such as the hydrogen bonds in TbAQP2 between Asn130 and the backbone carbonyl of 260 and the backbone amine of Ala132, the hydrogen bond between Ser263 and Gln155 and the hydrogen bond between Asn261 and the backbone amide of Ser263. However, both Asn residues in the NPS/NLA motif of PfAQP form hydrogen bonds with one of the glycerol molecules in the channel, whereas the Asn residues in the NPS/NSA motif in TbAQP2 form weak van der Waals interactions with the glycerol molecules, and Asn261 also makes a hydrogen bond to the backbone carbonyl of Leu129. The position of the intracellular loop between H2 and H3 is also conserved between TbAQP2 and PfAQP (*Figure 4c*) with the backbone carbonyls of residues 126, 127, and 128 aligning and all forming hydrogen bonds in both channels to glycerol molecules. The conservation in structure is reflected in the conserved position of the glycerol molecules in the intracellular half of the channel in TbAQP2 and PfAQP, although the orientation of the hydroxyl groups of the glycerol molecules differs. A divergence in the position of the extracellular loop between H6 and H7 in TbAQP2 and PfAQP (*Figure 4c*) leads to a wider channel in TbAQP2 and no ordered glycerol molecules are observed in the channel at this point, unlike in other aquaglycero-porins (*Figure 4d*). Another difference between the transport of glycerol by TbAQP2 and PfAQP is that there is a water molecule coordinated between each glycerol molecule in PfAQP, whereas in TbAQP2 the glycerol molecules interact directly. Any role of water molecules in the transport of glycerol by TbAQP2 remains to be elucidated as the resolution of the current cryo-EM structure is insufficient for resolving them.

## Molecular dynamics simulations show impaired pentamidine transport in mutants

To assess the dynamic properties of the structurally resolved pentamidine binding pose, all-atom molecular dynamics (MD) simulations were performed on TbAQP2 in a tetrameric state (see Methods). In these simulations, both the protein and pentamidine are very stable (protein RMSD = 0.17 ± 0.01 nm and pentamidine RMSD = 0.16 ± 0.03 nm, see *Figure 5—figure supplement 1a, e*) and pentamidine binding prevents water flux through the channel (*Figure 5a, b*). The stability of the pentamidine is enabled by a number of extremely high occupancy interactions with specific residues lining the TbAQP2 pore (*Figure 5—figure supplement 1b*). Hydrogen bond analysis reveals around 23 hydrogen bonds may be made between pentamidine and the channel, with 6 of these relatively stable (*Figure 5—figure supplement 2a, b*). In contrast, water shows a different pattern of high occupancy interactions in the pore, with MD snapshots highlighting several key positions of where waters interact with the channel as they flow through (*Figure 5—figure supplement 2c*). The presence of pentamidine both severely reduces the number of waters in the channel and slightly impacts their orientation (*Figure 5—figure supplement 2d, e*). It was not possible to perform MD simulations of TbAQP2 in the presence of melarsoprol, because the presence of an arsenic atom causes force field parameterisation issues.

We then explored the translocation of pentamidine through the TbAQP2 channel. The positively charged state of pentamidine suggests that it might be translocated by the $\Delta\Psi$ component of the plasma membrane potential. To test this, we applied a *z*-dimension electric field to simulations containing monomeric TbAQP2 in a membrane, using either a positive field (the opposite direction to the physiological membrane) or a physiologically correct negative field polarity. In all simulations, the resulting potential causes the pentamidine to translocate through TbAQP2, and in some simulations (one out of three repeats for each condition) the pentamidine fully leaves the channel (*Figure 5c*). For the physiologically correct negative field, the pentamidine moves downwards (*Figure 5c*), suggesting that the membrane potential is able to import pentamidine into the cell, consistent with experimental evidence that TbAQP2-mediated pentamidine uptake is sensitive to ionophores that depolarise the plasma membrane (*De Koning, 2001*; *Alghamdi et al., 2020*).

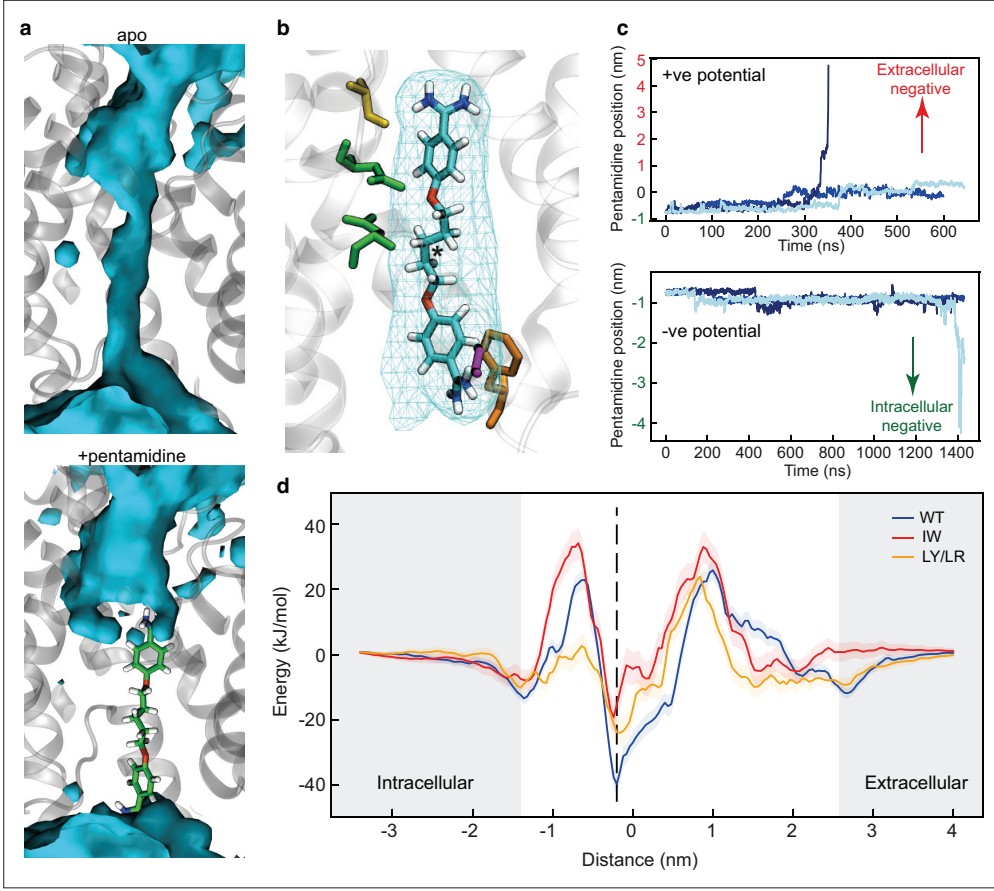

**Figure 5.** Molecular dynamics (MD) simulations of pentamidine-bound TbAQP2. (**a**) Density of water molecules (blue) through the TbAQP2 central cavity from MD. Bound pentamidine (sticks, green) abolishes the ability of TbAQP2 to transport water molecules. (**b**) View of pentamidine (cyan, carbon; red, oxygen; blue, nitrogen; white, hydrogen) bound to TbAQP2. Cyan mesh shows the density of the molecule across the MD simulation, and the asterisk shows the position of the centre of mass (COM). (**c**) Upon application of a membrane potential, the pentamidine position as defined by the COM moves along the z-dimension in relation to the COM of the channel, with three independent repeats shown in different shades of blue. The bottom graph is for the potential in the physiological direction (negative intracellular). (**d**) Energy landscapes for pentamidine through the TbAQP2 central cavity as calculated using umbrella sampling. Separate calculations were made for monomeric WT TbAQP2 and TbAQP2 with I110W (IW) or L258Y/L264R (LY/LR) mutations. Each trace is built from 167 × 40 ns windows, with the histogram overlap and convergence plotted in *Figure 5—figure supplement 3a, b*. The position of the membrane phosphates is shown as grey bars, and the structural binding pose is shown as a dotted line.

The online version of this article includes the following figure supplement(s) for figure 5:

**Figure supplement 1.** Molecular dynamics (MD) simulations.

**Figure supplement 2.** Water in molecular dynamics (MD) simulations.

**Figure supplement 3.** Molecular dynamics simulations.

## Drug-resistant mutations in TbAQP2

*Trypanosoma brucei* contains two related aquaglyceroporins, TbAQP2 and TbAQP3, with only TbAQP2 being able to transport pentamidine and melarsoprol into the trypanosome (*Baker et al., 2012*; *Munday et al., 2014*). Drug resistance has arisen through recombination between the two genes and the expression of TbAQP2–AQP3 chimeras (*Graf et al., 2015*; *Graf et al., 2013*; *Pyana Pati et al., 2014*). There are 68 amino acid residues that differ between TbAQP2 and TbAQP3, and the drug-resistant chimeras can contain 20–43 differences to TbAQP2 (*Figure 1*). A number of these residues have been mutated individually in attempts to identify key residues involved in drug translocation and thus potentially resistance. At the intramembrane ends of the two transmembrane half-helices H3 and H7 are the aforementioned NSA/NPS motifs, Asn130–Ser131–Ala132 and

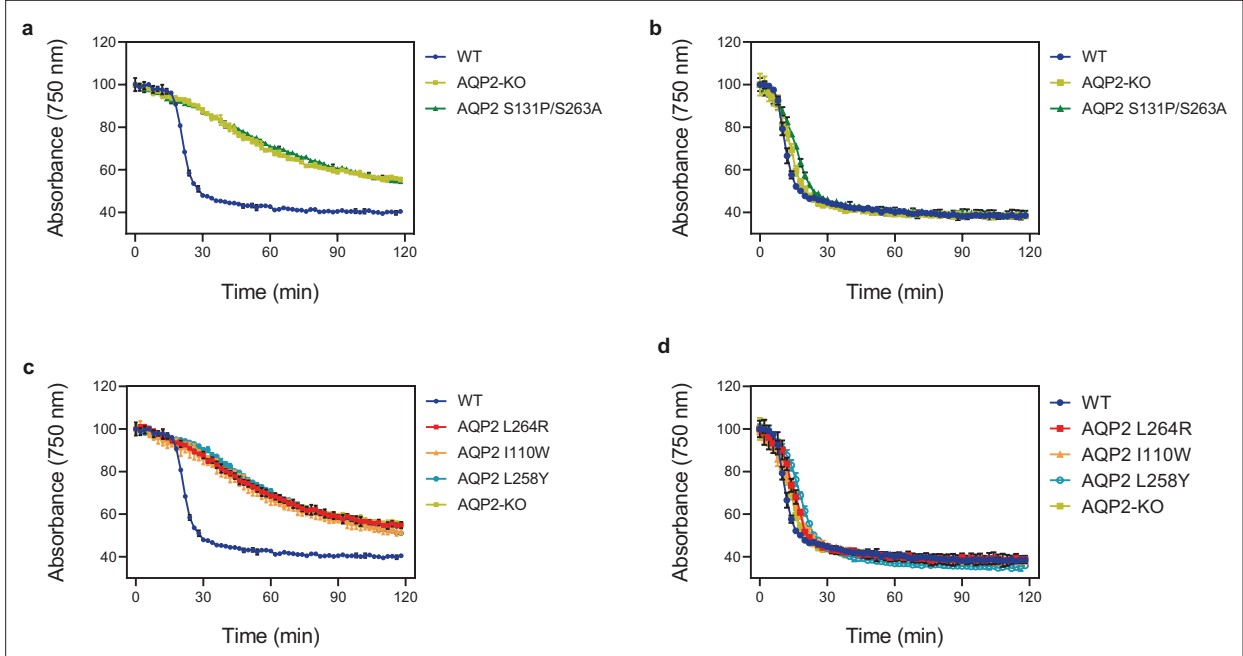

**Figure 6.** Lysis assay of *T. brucei* cells. Lysis assay with *T. b. brucei* wild-type and various AQP2 mutant cell lines treated with arsenical compounds: (**a, c**) cymelarsan; (**b, d**) phenylarsine oxide, a different arsenic-containing trypanocide that enters the parasite independently of TbAQP2 (**Munday et al., 2015b**; **De Koning, 2001**; **Munday et al., 2014**) and is thus used as a control for transporter-related arsenic resistance versus resistance to arsenic per se. The cells were placed in a cuvette and treated with either compound at *t* = 10 min. All points shown are the average of triplicate determinations and SD. When error bars are not visible, they fall within the symbol. The slow decline with cymelarsan over time in AQP2-KO and the mutant cell lines is attributable to residual uptake of the compound through the TbAT1/P2 transporter (**Carter and Fairlamb, 1993**; **de Koning et al., 2000**).

Asn261–Pro262–Ser263, respectively (**Figure 4a**). In aquaporins such as TbAQP3, these conserved residues are the canonical NPA/NPA motifs, and mutations were made (S131P + S263A) to convert the NSA/NPS motif in TbAQP2 to NPA/NPA. This resulted in a 95% reduction in pentamidine transport by the mutant, although this was still measurably higher than that observed in a TbAQP2 knockout cell line (**Alghamdi et al., 2020**) and a similar reduction in the transport of the melarsoprol analogue cymelarsan (**Figure 6**). The cryo-EM structures show that both Ser131 and Ser263 are critical in stabilising this region of the transporter through specific hydrogen bonds with Gln155 and the backbone amine group of Ala132 (**Figure 4a**) and do not make any direct interactions either to the drugs or the glycerol molecules. Thus, the decrease in pentamidine uptake observed in the S131P + S263A TbAQP2 mutant is probably more complex than previously thought, as it seems unlikely to have arisen through simple changes in transporter–drug interactions or a narrowing of the channel.

Based on a previous modelling study, several other mutations were predicted to potentially affect drug uptake in the TbAQP2–TbAQP3 chimeras (**Alghamdi et al., 2020**). Based on the cryo-EM structures, the mutated residues fall into three distinct groups: those not in the channel (I190T and W192G), those lining the channel and within van der Waals distance of pentamidine (I110W, L258Y, and L264R) and those lining the channel, >3.9 Å from pentamidine and observed to make contacts to pentamidine during MD simulations (L84W/M, L118W/M, and L218W/M). In all instances, there was a >90% decrease in pentamidine uptake and in many cases no residual pentamidine transport was detected (e.g. I110W, L264R, and L84W) (**Alghamdi et al., 2020**). To understand the molecular basis for the dramatic decreases observed in pentamidine transport, we performed MD simulations on a subset of the mutants.

Three mutations were chosen for MD simulations (I110W, L258Y, and L264R), all of which were defective in uptake of cymelarsan (**Figure 6**) as well as pentamidine (**Alghamdi et al., 2020**). These three residues all make high occupancy contacts to pentamidine in the MD simulations (**Figure 5— figure supplement 1b**) and residues Ile110 and Leu264 make van der Waals contacts with pentamidine in the cryo-EM structure (Leu258 is 5.5 Å away from pentamidine). The double mutant L258Y + L264R was used in the MD simulations as this is found in a number of drug-resistant *T. brucei*

strains such as P1000 and Lister 427MR (*Munday et al., 2014*). The mutant I110W is found in the *T. brucei* drug-resistant strain R25 (*Unciti-Broceta et al., 2015*), but in the absence of the L258Y + L264R double mutation (*Figure 1*). Therefore, we performed atomistic MD simulations of monomeric TbAQP2 with either the I110W (hereafter IW) or L258Y + L264R (hereafter LY/LR) mutations present, as well as with wild-type monomeric TbAQP2. Each protein was simulated both with pentamidine bound and removed (apo). The mutations do not affect TbAQP2 stability (*Figure 2—figure supplement 6e*), but do destabilise pentamidine binding, causing it to shift away from the initial binding pose along the z-axis (*Figure 5—figure supplement 1c*), suggesting a lowered affinity for the structural pose. The pore radii are unchanged by the mutations (*Figure 5—figure supplement 1d*), meaning that this effect is likely to be a loss of specific interactions rather than a more general effect on protein structure.

We next computed energy landscapes for pentamidine moving along the z-axis through the channel using potential of mean force (PMF) calculations with umbrella sampling. The data for WT TbAQP2 reveals a clear energy well for the structurally resolved pentamidine at –0.2 nm (*Figure 5d*, *Figure 5—figure supplement 3b*). The depth of this well (–40 kJ/mol) suggests a very high-affinity binding site, consistent with previous kinetic analyses (*Bridges et al., 2007*; *De Koning, 2001*; *Teka et al., 2011*), and is much higher than previously investigated docked poses (*Alghamdi et al., 2020*) of pentamidine in models of TbAQP2. Notably, this energy well is much shallower for both the IW (–20 kJ/mol) and LY/LR mutants (–25 kJ/mol), representing a relative reduction in binding likelihood of about 2800- and 400-fold, respectively. In biochemical terms, these changes would result in a $K_m$ shift from ~50 nM to 50 or 300 µM, respectively, all but abolishing uptake at pharmacologically relevant concentrations; the experimental $K_m$ for pentamidine on wild type TbAQP2 is 36 nM (*De Koning, 2001*). The reduction in pentamidine binding likelihood is consistent with the increased z-axis dynamics seen in *Figure 5—figure supplement 1c*. Flanking the central energy well are large energy barriers at about –0.7 and 1 nm (*Figure 5d*), similar to those seen for urea translocation in HsAQP1 (*Hub and Groot, 2008*). These would presumably slow pentamidine entry into and through the channel, and previous studies did reveal a slow transport rate for pentamidine (*De Koning, 2001*; *Teka et al., 2011*). We note that previous studies using these approaches saw energy barriers of a similar size, and that these are reduced in the presence of a membrane voltage (*Alghamdi et al., 2020*; *Chen et al., 2024*). The energy barriers are highest in the IW mutant, and interestingly, the inner (–0.7 nm) barrier is lower in the LY/LR mutant (*Figure 5—figure supplement 3e*). In all cases, especially WT, small energy minima are observed at about –1.35 and +2.7 nm; these represent the drug binding at the entrance and exit of the channel (*Figure 5—figure supplement 3d*). The differences in landscape between the WT and mutants help explain how these mutations confer a resistance to pentamidine, mostly through reduced binding affinity.

## Discussion

It has only recently become accepted that the aquaglyceroporin AQP2 in *T. brucei* is acting as a *bona fide* transporter of pentamidine and melarsoprol that is essential for translocation of the drugs into the cell (*Alghamdi et al., 2020*). The cryo-EM structures of TbAQP2 identify the binding site for melarsoprol and pentamidine in a wide channel that has, at least for part of the trajectory, similar dimensions to channels in other aquaglyceroporins (*Figure 4d*). The overall similarity between TbAQP2 and the structures of other aquaglyceroporins suggests a similarity in the mechanism of glycerol transport, despite differences in the amino acid residues lining the channel. It also suggests that there is scope for pharmacological targeting of other aquaglyceroporins, either for uptake of cytotoxic drugs or inhibition of glycerol uptake. Leishmaniasis, for instance, is another neglected tropical disease, caused by parasitic protozoa of the genus *Leishmania*; it is commonly treated by pentavalent antimonials meglumine antimoniate and sodium stibogluconate. The antimony that is released in the acidic parasitophorous vacuole enters the parasite through its AQP1 aquaporin (*Plourde et al., 2015*). Widespread antimonial drug resistance in *Leishmania donovani* in India has been linked to a frameshift mutation in LdAQP1 (*Imamura et al., 2016*). In another example, *Plasmodium* spp. contain a single aquaglyceroporin and studies in mouse models have shown that a knockout of the aquaglyceroporin leads to a significant loss of virulence and reduction in erythrocyte infectivity by merozoites (*Promeneur et al., 2018*), suggesting that a specific AQP inhibitor could be part of antimalarial treatments. Interestingly, pentamidine is a known antimalarial (*Bray et al., 2003*) and although a substrate of the *P. falciparum*

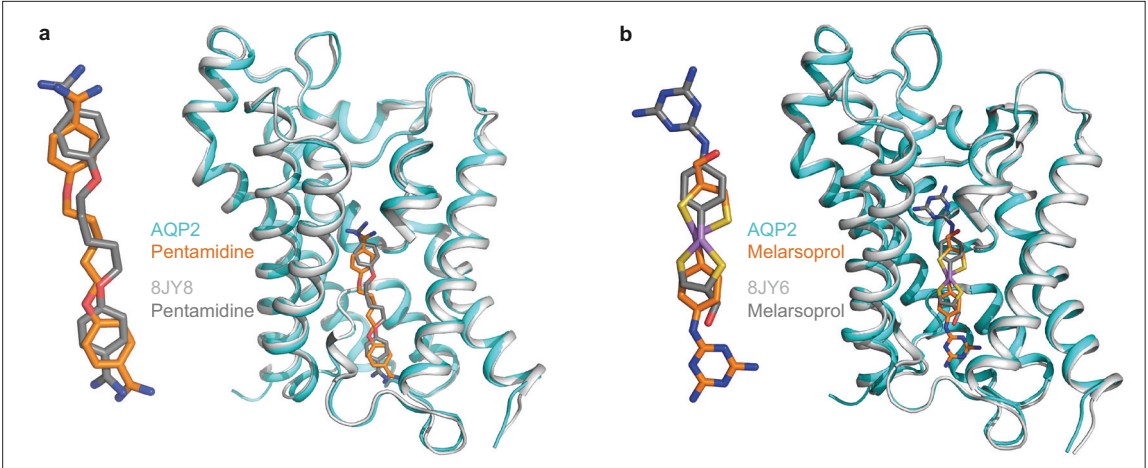

**Figure 7.** Comparison between drug-bound AQP2 structures. Superposition of an AQP2 protomer from this work (cyan) with a protomer determined by *Chen et al., 2024* (grey), with the positions of drugs shown in stick representation: (**a**) pentamidine and (**b**) melarsoprol.

choline transporter, does not act through inhibition of that carrier (*Biagini et al., 2004*); inhibition/permeation of PfAQP by pentamidine has not been investigated to date. The structure determination of TbAQP2 highlights new opportunities for drug development targeting the aquaporin family to treat a number of parasitic diseases whose current treatments often suffer from severe side effects and growing problems in drug resistance. A thorough understanding of the interaction of extended, flexible drugs like pentamidine and melarsoprol with AQP channel linings could contribute to the design of new substrates able to penetrate not just protozoan but also human AQPs.

After this work was completed, a paper was published describing the structures of TbAQP2 bound to pentamidine and melarsoprol (*Chen et al., 2024*). The structures of TbAQP2 are very similar to the ones described here (RMSD 0.6 Å), despite the environments of the purified channels being significantly different. In the work described here, TbAQP2 was purified in detergent and then incorporated into saposin A (SapA) (*Frauenfeld et al., 2016*) lipid nanoparticles (see Methods), whereas (*Chen et al., 2024*) incorporated TbAQP2 into lipid nanodiscs formed from a helical membrane scaffold protein (*Bayburt et al., 2002*; *Denisov et al., 2004*). Lipid nanodiscs and nanoparticles are considered better mimetics of a lipid bilayer than detergent micelles, although biophysical studies indicate that there are significant differences between these lipidic structures and a biological membrane (*Stepien et al., 2020*; *Denisov et al., 2005*). Comparison of the pentamidine in the different structures indicates good agreement in its position within the channel, although the pose of the drug does differ slightly (*Figure 7a*). In contrast, there is a significant difference in the position and pose of melarsoprol between the different structures (*Figure 7b*). Melarsoprol in the TbAQP2 structure from *Chen et al., 2024* is shifted further towards the extracellular surface of the channel and is rotated approximately 180° about an axis in the membrane plane and centred on the arsenic atom (*Figure 7b*). This unexpected difference in orientation of melarsoprol in the two structures may perhaps not be so surprising as could be initially thought. The two structures represent the thermodynamically most stable states under the buffer conditions and detergent/lipid nanoparticles used for protein purification, which were different. In addition, the site of localisation of the drugs in the TbAQP2 channel is not a physiologically relevant 'binding site' as used to describe, for example, the orthosteric binding site in G protein coupled receptors where a hormone or neurotransmitter binds. Thus, evolution has not evolved a high-affinity binding site in the TbAQP2 channel and neither were the drugs designed to bind to TbAQP2. The channel merely facilitates the entry of the drugs into the cell through passive diffusion and it is not necessary for its mode of action to bind to TbAQP2 in a specific pose. Indeed, a priori, there is no reason to think that melarsoprol has to traverse the channel in any particular orientation, although under cellular conditions where a membrane potential is present, it might be anticipated that the melamine moiety may enter the channel first given its net positive charge. Regrettably, in the absence of MD simulations (due to the issues with parameterisation of the arsenic atom), these ideas have to remain untested.

# Materials and methods

## Data reporting

Sample size was not predetermined by statistical methods, no randomisation was applied to experiments and investigators were not blinded to experimental allocation or outcome assessment.

## Cloning and expression of wild-type AQP2

A construct encoding wild-type *T. brucei* AQP2 (residues 1–312) that included a tobacco etch virus (TEV) cleavage site, eGFP, and a decahistidine tag (TbAQP2-TEV-eGFP-His10) was generated and cloned into a pFastBac vector (Thermo Fisher) using standard ligation procedures. Recombinant pFastBac-TbAQP2-TEV-eGFP-His10 vector was transformed into Stellar Competent Cells (Takara) following the Bac-to-Bac Baculovirus Expression System manual (Thermo Fisher). Expression of AQP2 in *Spodoptera frugiperda* 9 (Sf9) cells (ATCC CRL-1711) was performed following the Bac-to-Bac Baculovirus Expression System manual (Thermo Fisher). Sf9 cells were not tested for mycoplasma by the users (scientifically unnecessary as the cells were used solely for protein production and mycoplasma cannot alter the channel's structure). The cells were then collected by centrifugation (10 min, 4°C, 3000 × *g*), flash-frozen in liquid nitrogen, and stored at –80°C until further use.

## Purification of AQP2 bound to either glycerol, pentamidine, or melarsoprol and reconstitution into saposin-A nanoparticles (SapA-NP)

All steps described below were carried out at 4°C. *Sf*9 cells were defrosted on ice, resuspended in cold homogenisation buffer (100 mM NaCl, 20 mM HEPES pH 7.5, 5% (vol/vol) glycerol, 2 mM PMSF, 1× Complete Protease Inhibitor Cocktail (Sigma)) and homogenised by applying 20 strokes in a Potter-Elvehjem glass homogeniser on ice. Membranes were solubilised using 1% (wt/vol) final concentration *n*-dodecyl-β-ᴅ-maltopyranoside (DDM, Anatrace) and a final concentration of 0.03% (wt/vol) cholesteryl hemisuccinate (CHS, Anatrace) for 4 hr at 4°C on a roller platform. Insoluble material was removed by ultracentrifugation (100,000 × *g*, 60 min, 4°C). The solubilisate was supplemented with 20 mM imidazole (final concentration) and incubated (16 hr, 4°C) with 2 ml of $Ni^{2+}$-NTA agarose resin (QIAGEN) per litre of *Sf*9 cell culture. The resin was then applied to a gravity-flow column and washed three times with 20 column volumes washing buffer (100 mM NaCl, 20 mM HEPES pH 7.5, 2% (vol/vol) glycerol, 2 mM PMSF, 1× Complete Protease Inhibitor Cocktail (Sigma), 0.05% (wt/vol) DDM, 0.03% (wt/vol) CHS, 100 mM imidazole) and was eluted twice with one column volume elution buffer (100 mM NaCl, 20 mM HEPES pH 7.5, 2% (vol/vol) glycerol, 1× Complete Protease Inhibitor Cocktail (Sigma), 2 mM PMSF, 500 mM imidazole) each. Both eluates were combined and concentrated in an Amicon Ultra 4 ml centrifugal filter (100,000 MWCO, Merck) and aggregates were removed using a syringe-driven filter unit (Millex-GV, 0.22 μm, PVDF, 13 mm, Merck). The filtrate was loaded at 0.4 ml/min onto a Superdex Increase S200 10/300 (GE Healthcare) size exclusion chromatography (SEC) column, pre-equilibrated with 0.05% (wt/vol) DDM, 100 mM NaCl, 20 mM HEPES pH 7.5. The peak fractions containing AQP2 were pooled and concentrated as above. Protein concentration was estimated by the Pierce BCS assay according to the manufacturer's instructions.

SapA-NPs were prepared and stored in SapA-NP buffer consisting of 150 mM NaCl, 50 mM sodium acetate pH 4.8 (*Chien et al., 2017*). Purified TbAQP2 tetramer was combined at a TbAQP2:SapA:lipid molar ratio of 0.1:1:10 and dialysed (20 hr, 4°C) against SapA-reconstitution buffer (100 mM NaCl, 20 mM HEPES pH 7.5). SapA-NPs, unincorporated TbAQP2 and SapA were removed by SEC and SapA-TbAQP2-NP tetramer complex was concentrated using Amicon Ultra centrifugal filters (100,000 MWCO; Merck). Melarsoprol or pentamidine was added to the purified SapA-TbAQP2-NP tetramer complex (final concentration 1 mM, 2 hr, 4°C) applied to a Superdex Increase S200 10/300 SEC column (GE Healthcare) and fractions containing the complex were concentrated using Amicon Ultra centrifugal filters (100,000 MWCO; Merck) to final concentrations between 0.8 and 1.6 mg/ml. Final samples were centrifuged (1 hr, 18,000 × *g*, 4°C) immediately prior to cryo-EM grid preparation.

## Vitrified sample preparation and data collection

Cryo-EM grids were prepared by applying 2 μl of purified SapA-TbAQP2-NP with either glycerol, pentamidine or melarsoprol bound (0.8–1.6 mg/ml) onto either glow-discharged PEGylated UltrAuFoil R1.2/1.3 300 Mesh Gold grids (Agar scientific, No. AGS187-3) or glow-discharged Quantifoil

R1.2/1.3 300 Mesh Gold grids (Agar scientific, No. AGS143-8). Grids were blotted with filter paper for 4 s before plunge-freezing in liquid ethane (at −181°C) using a Thermo Fisher Scientific Vitrobot Mark IV (95% relative humidity, 4°C). All cryo-EM datasets were collected on Titan Krios microscopes (Thermo Fisher Scientific) operating at 300 kV. SapA-TbAQP2-NP with glycerol was collected at the Nanoscience Centre (University of Cambridge) with a Falcon 3 detector to obtain dose-fractioned movies. 2881 movies were recorded with ×75,000 magnification at 1.07 Å per pixel using an exposure time of 60 s resulting in a total exposure of (30 e/pixel) and a target defocus range of −1.4 and −3.2 μm. Datasets of SapA-TbAQP2-NP with bound pentamidine or melarsoprol were collected at the cryo-EM facility at the Department of Biochemistry, University of Cambridge; images were recorded using a K2 detector in counting mode, operating with GIF Quantum LS imaging filter (Gatan) with a slit width of 20 eV to obtain dose-fractioned movies with a 100-μm objective aperture. 2400 movies were recorded for SapA-TbAQP2-NP with pentamidine bound and 3000 movies for melarsoprol bound complex. Micrographs for both datasets were recorded at a magnification of ×130,000 (1.07 Å per pixel) as dose-fractioned movie frames with an exposure time of 15 and 12 s, respectively, resulting in a total exposure of 54 and 57.5 e/Å$^2$ with target defocus ranges of −1.5 and −3.3 μm, and −1.2 and −3 μm, respectively.

## Cryo-EM data processing and 3D reconstruction

Cryo-EM data processing was conducted inside the Scipion platform (*de la Rosa-Trevín et al., 2016*). Image stacks (2881 glycerol-bound, 4086 melarsoprol-bound, and 3262 pentamidine-bound) were subjected to beam-induced motion correction using Relion's implementation of MotionCor2 (*Zheng et al., 2017*) by dividing each frame into 5 × 5 patches. CTF parameters were estimated from dose-weighted micrographs as follows: estimations made with GCTF (*Zhang, 2016*) with equiphase averaging and with CTFFIND-4.1 (*Sorzano et al., 2021*; *Rohou and Grigorieff, 2015*) were compared using «xmipp-ctf consensus». Outputs from these consensus (micrographs for which the two algorithms agreed in their estimations) underwent an additional round of ctf estimation with «xmipp-ctf estimation» (*Sorzano et al., 2007*), followed by a second round of xmipp-ctf consensus for micrograph curation based on defocus, astigmatism, resolution and ice thickness. As a result, the number of micrographs for downstream analysis was 2102, 3586 and 2983 for SapA-TbAQP2-NP bound to either glycerol, melarsoprol, or pentamidine, respectively.

Autopicking was performed using crYOLO (*Wagner et al., 2019*), that yielded 849,607 particles, 1,927,102 particles and 1,263,534 particles for SapA-TbAQP2-NP bound to either glycerol, melarsoprol, or pentamidine, respectively. Particles were extracted in a box size of 250 px at the original sampling rate of the micrographs (1.07 Å/px) and subjected to several rounds of 2D classifications with cryoSPARC (*Punjani et al., 2017*). To further curate the particle set (ligand-free: 100,075; melarsoprol: 202,752; pentamidine: 114,206). Particles were re-extracted with a box of 320 px before 3D classifications.

For the glycerol-bound dataset, an ab initio 3D model was generated using stochastic gradient descent algorithm implemented in cryoSPARC, followed by non-uniform refinement (*Punjani et al., 2020*) with C4 symmetry and a static mask encompassing the transmembrane domains and nanodiscs belt, while excluding the mobile GFP tags. With these settings, a 3.2-Å resolution map was obtained. For the melarsoprol and pentamidine datasets, six ab initio models were generated. Particles from the best classes (melarsoprol: 126,551; pentamidine: 83,845) were used for non-uniform refinement with the same strategy as for the glycerol-bound particles, yielding a 3.2- and 3.7-Å resolution map, respectively.

The global resolution was calculated following the FSC gold standard threshold of 0.143. The refined maps were sharpened using deepEMhancer (*Sanchez-Garcia et al., 2021*) to facilitate model building. Finally, voxel size for melarsoprol and pentamidine was re-adjusted to 1.042 Å/px using Relion post-process (*Zivanov et al., 2018*). Local resolution was estimated with MonoRes (*Vilas et al., 2018*).

## Structure determination and model refinement

The monomer structure of TbAQP2 was predicted using the AlphaFold2 program as a starting point for further refinement (*Jumper et al., 2021*). The final cryo-EM map was converted to an mtz file with a range of sharpening and blurring options using CCPEM (*Wood et al., 2015*; *Burnley et al., 2017*).

EMDA was used for fitting maps (*Warshamanage et al., 2022*). Real-space refinement of the atomic coordinates into the cryo-EM map was performed using Phenix (*Liebschner et al., 2019*) and Coot software (*Emsley et al., 2010*). Finally, models in the asymmetric unit were refined against unsharpened and unweighted half maps using Servalcat Refmac5 pipeline (*Yamashita et al., 2021*) with *C*4 symmetry constraints. The reference structure restraints were prepared with ProSmart (*Nicholls et al., 2014*) using AlphaFold2 predicted models from the AlphaFold DB (*Varadi and Velankar, 2023*). The final models were evaluated for geometry, close contacts, and bond parameters using MolProbity (*Chen et al., 2010*). Graphic representations of the fitted coordinates into electron density and the final cryo-EM map were generated using PyMOL (*DeLano, 2002*) and UCSF ChimeraX (*Meng et al., 2023*; *Pettersen et al., 2021*) packages.

## MD simulations

The model for MD simulations was based on the structure of TbAQP2 with pentamidine bound and was built into simulation boxes using CHARMM-GUI (*Jo et al., 2007*; *Lee et al., 2016*) (see *Supplementary file 1* for a list of systems built). Protein atoms were described with the CHARMM36m force field (*Best et al., 2012*; *Huang et al., 2017*) TbAQP2 was built either as a tetramer or a monomer. Side chain p$K_a$s were assessed using propKa3.1 and side chain side charge states were set to their default (*Søndergaard et al., 2011*). The pentamidine molecule used existing parameters available in the CHARMM36 database under the name PNTM with a charge state of +2 to reflect the predicted pKas of >10 for these groups (*Wang et al., 2010*) and in line with previous MD studies (*Alghamdi et al., 2020*). For the ligand-free simulations, the pentamidine molecule was removed prior to simulation.

The proteins were built into membranes comprising 20% cholesterol and 80% POPC, solvated with explicit water molecules using the TIP3P model, and neutralised with $K^+$ and $Cl^-$ to 150 mM. Box sizes were initially set to 10 × 10 × 10 nm for the tetrameric systems and 6.5 × 6.5 × 10 nm for the monomeric systems. The use of an all-atom fixed charge force field with explicit solvent and membrane is optimal to understanding the protein–drug interactions central to this study. Each system was minimised and equilibrated according to the standard CHARMM-GUI protocol (*Jo et al., 2007*; *Lee et al., 2016*). Production simulations using random seeds were run in the NPT ensemble with 2 fs time steps, temperatures held at 303.5 K using a velocity-rescale thermostat and a coupling constant of 1 ps, and pressure maintained at 1 bar using a semi-isotropic Parrinello–Rahman pressure coupling with a coupling constant of 5 ps (*Bussi et al., 2007*; *Parrinello and Rahman, 1981*). Short-range van der Waals and electrostatics were cut off at 1.2 nm, and the particle mesh Ewald method was used for long-range Lennard–Jones interactions (*Darden et al., 1993*). Bonds containing hydrogen were constrained using the LINCS algorithm.

Membrane potential simulations were run using the computational electrophysiology protocol. An electric field of 0.1 V/nm was applied in the *z*-axis dimension only, to create a membrane potential of about 1 V (see *Figure 5—figure supplement 3a*). Note that this is higher than the physiological value of 87.1 ± 2.1 mV at pH 7.3 in bloodstream *T. brucei* (*de Koning and Jarvis, 1997*) and was chosen to improve the sampling efficiency of the simulations. The protein and lipid molecules were visually confirmed to be unaffected by this voltage, which we quantify using RMSF analysis on pentamidine-contacting residues (*Figure 5—figure supplement 3b*).

PMF calculations were built from initial steered MD pulls using an umbrella potential along a coordinate of the COM of the pentamidine molecule and the COM of select residues in TbAQP2 which form the pentamidine binding site (Leu84, Ile110, Val114, Leu122, Gly127, Leu129, Leu218, Val222, Phe226, Leu244, Val245, Ala259, Met260, and Leu264). Pulls were run in the *z*-axis in either a positive or negative direction, at a rate of 1 nm per ns and with a force constant of 1000 kJ/mol/nm². Snapshots were taken along the coordinates every 0.05 nm, using a script from Dr. Owen Vickery (https://doi.org/10.5281/zenodo.3592318), with a total of 167 snapshots to construct the reaction coordinate. End points for the coordinate were determined based on the pentamidine no longer making any contacts (<0.4 nm) with any atoms from the protein or membrane. Simulations were run for 40 ns per window with a 5000 kJ/mol/nm² umbrella restraint imposed to keep the system in place with respect to the reaction coordinate. An atom in Leu218 was chosen as a reference for the treatment of periodic boundary conditions. Forces were written every 1 ps. PMF profiles were constructed using the weighted histogram analysis method (WHAM) (*Kumar et al., 1992*) implemented in *gmx wham* (*Hub et al., 2010*) between –3.5 and +4.5 nm of the coordinate and with cyclisation applied. The first 5 ns

of each window was discarded as equilibration time, and 200 Bayesian bootstraps were used to estimate the PMF error. Adequate coverage of the landscape was assessed based on histogram overlap (*Figure 5—figure supplement 2c*), and convergence was determined based on re-running the WHAM calculations on longer windows until the landscapes converged (*Figure 5—figure supplement 2d*).

All simulations were run in Gromacs 2020.3 (https://doi.org/10.5281/zenodo.7323409) (*Abraham et al., 2015*). Data were analysed using Gromacs tools and VMD (*Humphrey et al., 1996*). Pentamidine interactions with TbAQP2 were assessed using PyLipID (*Song et al., 2022*). Pore radii were assessed using the CHAP package (*Klesse et al., 2019*). Plots were made using Matplotlib (*Hunter, 2007*) or Prism 9 (GraphPad). MD input and output coordinate files are available at https://osf.io/zc235/.

### Docking and hotspots experiments

Fragment Hotspot Maps (FHMs) is a powerful tool for predicting the binding hotspots of small molecules on protein surfaces (*Radoux et al., 2016*). FHMs are based on the concept that certain chemical fragments are more likely to interact with specific regions of a protein's surface, or 'hotspots', than others. By systematically mapping the interactions between a library of fragment molecules and a target protein, FHMs can identify key hotspots and help guide the design of new small molecules with improved binding affinity and specificity. The development of FHMs has been driven by advances in computational methods for predicting protein–ligand interactions, as well as by improvements in experimental techniques for mapping the 3D structure of protein–ligand complexes. FHMs have been used to guide drug discovery efforts in a variety of therapeutic areas, including oncology, infectious diseases, and neurology.

One key advantage of FHMs is that they can help to identify 'druggable' regions of a protein's surface that may be otherwise difficult to target with small molecules. For example, many protein–protein interaction interfaces are large and flat, making it challenging to design small molecules that can disrupt these interactions. FHMs can help to identify the specific regions of the interface that are most important for binding and guide the design of molecules that can effectively target these regions. Another advantage of FHMs is that they can be used to predict the binding modes of small molecules that have not yet been synthesised or tested experimentally. This can be particularly useful in early-stage drug discovery, where computational methods are often used to screen large databases of compounds for potential hits. Overall, FHMs represent a powerful approach for predicting and optimising small molecule binding to protein surfaces.

We used FHMs program to generate hotspots for AQP2 monomer and probe protein's potential ligand binding sites. The superposition of the final structures and the predicted hotspot maps shows a clear overlap between the hotspots and the two drugs (*Figure 2—figure supplement 1d–f*).

### Lysis assay with arsenical drugs

The effects of cymelarsan and phenylarsine oxide (PAO) on *T. brucei* Lister 427 bloodstream forms were measured as the decrease in cell absorbance at $\lambda$ 750 nm over time, based on the reduction of cell motility and increased cell lysis leading to reduced light scatter and absorbance in the cuvette (*Fairlamb et al., 1992*), exactly as described (*Ungogo et al., 2022*). Briefly, $2 \times 10^6$ cells in 200 µl buffer were placed in the wells of a 96-well plate and absorbance recorded in a PHERAstar plate reader (BMG Labtech, Durham, NC, USA) for 10 min before the addition of 20 µl of 100 µM of either cymelarsan or PAO in aqueous buffer, or buffer alone.

### Acknowledgements

SNW's lab was funded by a Sir Henry Dale fellowship from the Wellcome Trust and the Royal Society London (Grant Number 101234/Z/13/Z) and by the Isaac Newton Trust Cambridge (Grant Number 15.40(a)). The work in CGT's laboratory was supported by core funding from the Medical Research Council (MRC U105197215). TS was supported by an MRC-DTP PhD fellowship, a Cambridge Trust Vice-Chancellor's Award, the Sackler fund and a King's College Cambridge PhD overrun fund. RAC, MSPS, and PJS are supported by Wellcome (208361/Z/17/Z). Aquaporin-related work in the HPdK lab was supported by the Medical Research Council (Grant Number G0701258) and a BBSRC Impact Accelerator award (BB/S506734/1). RW was supported by the Wellcome Trust (208398/Z/17/Z). MC is a Wellcome Investigator (217138/Z/19/Z). This project made use of time on ARCHER2 granted

via the UK High-End Computing Consortium for Biomolecular Simulation, HECBioSim (http://hecbi-osim.ac.uk), supported by EPSRC (EP/R029407/1). In addition, simulations were performed using the computational facilities of the Advanced Computing Research Centre, University of Bristol (http://www.bristol.ac.uk/acrc/). This work benefited from access to the Instruct Image Processing Centre (I2PC), an Instruct-ERIC centre; financial support was provided by PID12540. For the purpose of open access, the MRC Laboratory of Molecular Biology has applied a CC BY public copyright licence to any Author Accepted Manuscript version arising.

## Additional information

### Funding

| Funder | Grant reference number | Author |
|--------|------------------------|--------|
| Wellcome Trust | 10.35802/101234 | Simone Weyand |
| Isaac Newton Trust | 15.40(a) | Simone Weyand |
| Medical Research Council | U105197215 | Christopher G Tate |
| Medical Research Council | MRC-DTP PhD fellowship | Teresa Sprenger |
| Gates Cambridge Trust | Vice-Chancellor's Award | Teresa Sprenger |
| Wellcome Trust | 10.35802/208361 | Robin A Corey Phillip J Stansfeld Mark SP Sansom |
| Medical Research Council | G0701258 | Harry P De Koning |
| Biotechnology and Biological Sciences Research Council | Impact Accelerator Award BB/S506734/1 | Harry P De Koning |
| Wellcome Trust | 208398/Z/17/Z | Rangana Warshamanage |
| Wellcome Trust | 10.35802/217138 | Mark Carrington |

The funders had no role in study design, data collection, and interpretation, or the decision to submit the work for publication. For the purpose of Open Access, the authors have applied a CC BY public copyright license to any Author Accepted Manuscript version arising from this submission.

### Author contributions

Modestas Matusevicius, Investigation, Writing – review and editing, Built the models and performed refinements; Robin A Corey, Formal analysis, Investigation, Writing – original draft, Writing – review and editing, Performed the molecular dynamics simulations and analysed the MD simulation trajectories; Marcos Gragera, Investigation, Writing – review and editing, Processed the raw cryo-EM data; Keitaro Yamashita, Formal analysis, Investigation, Writing – review and editing, Identified and fitted the substrates and performed refinements; Teresa Sprenger, Investigation, Writing – review and editing, Cloned, overexpressed, purified, and analysed protein samples; Marzuq Ungogo, Investigation, Writing – review and editing, Performed cymelarsan uptake experiments; James N Blaza, Investigation, Writing – review and editing, Prepared grids, screened and collected data; Pablo Castro-Hartmann, Investigation, Writing – review and editing, Prepared grids, screened and collected data; Dimitri Y Chirgadze, Investigation, Writing – review and editing, Prepared grids, screened and collected data; Sundeep Chaitanya Vedithi, Investigation, Writing – review and editing, Performed hot spot analysis; Pavel Afanasyev, Formal analysis, Investigation, Writing – review and editing, Built the models and performed refinements; Roberto Melero, Investigation, Writing – review and editing, Processed the raw cryo-EM data; Rangana Warshamanage, Investigation, Writing – review and editing, Identified and fitted the substrates and performed refinements; Anastasiia Gusach, Formal analysis, Investigation, Writing – review and editing, Performed structural analyses; José-Maria Carazo, Formal analysis, Supervision, Writing – review and editing, Advised on cryo-EM image processing; Mark Carrington, Conceptualization, Formal analysis, Supervision, Writing – review and editing, Conceived

the project and advised on the mutational studies; Tom Blundell, Formal analysis, Supervision, Writing – review and editing, Advised on the hot spot analysis; Garib N Murshudov, Formal analysis, Supervision, Writing – review and editing, Advised on the refinement process; Phillip J Stansfeld, Formal analysis, Supervision, Writing – review and editing, Analysed MD simulation trajectories; Mark SP Sansom, Formal analysis, Supervision, Writing – review and editing, Analysed MD simulation trajectories; Harry P De Koning, Formal analysis, Supervision, Funding acquisition, Writing – original draft, Writing – review and editing, Advised on the mutational studies; Christopher G Tate, Formal analysis, Methodology, Writing – original draft, Writing – review and editing, Carried out structure analysis; Simone Weyand, Conceptualization, Resources, Supervision, Funding acquisition, Writing – original draft, Project administration, Writing – review and editing, Conceived the project, cloned, overexpressed, purified and analysed protein samples and managed the overall project

## Author ORCIDs
Robin A Corey ⓘ https://orcid.org/0000-0003-1820-7993
Marcos Gragera ⓘ https://orcid.org/0000-0002-3097-1743
James N Blaza ⓘ https://orcid.org/0000-0001-5420-2116
Dimitri Y Chirgadze ⓘ https://orcid.org/0000-0001-9942-0993
Pavel Afanasyev ⓘ https://orcid.org/0000-0002-6353-6895
José-Maria Carazo ⓘ https://orcid.org/0000-0003-0788-8447
Mark Carrington ⓘ https://orcid.org/0000-0002-6435-7266
Mark SP Sansom ⓘ https://orcid.org/0000-0001-6360-7959
Harry P De Koning ⓘ https://orcid.org/0000-0002-9963-1827
Christopher G Tate ⓘ https://orcid.org/0000-0002-2008-9183
Simone Weyand ⓘ https://orcid.org/0000-0002-7965-0895

Reviewer #1 (Public review): https://doi.org/10.7554/eLife.107460.3.sa1
Reviewer #2 (Public review): https://doi.org/10.7554/eLife.107460.3.sa2
Reviewer #3 (Public review): https://doi.org/10.7554/eLife.107460.3.sa3
Author response https://doi.org/10.7554/eLife.107460.3.sa4

# Additional files

## Supplementary files
MDAR checklist

Supplementary file 1. Details of molecular dynamics simulations on TbAQP2.

## Data availability
Structures have been deposited in the Protein Data Bank (PDB; https://www.rcsb.org/), and the associated cryo-EM data have been deposited in the Electron Microscopy Data Bank (EMDB; https://www.ebi.ac.uk/pdbe/emdb/) and the Electron Microscopy Public Image Archive (EMPIAR; https://www.ebi.ac.uk/empiar/): SapA-TbAQP2-NP bound to glycerol (8OFZ, EMD-16864, and EMPIAR-11410); SapA-TbAQP2-NP bound to pentamidine (8OFY, EMD-16863, and EMPIAR-11412); SapA-TbAQP2-NP bound to melarsoprol (8OFX, EMD-16862, and EMPIAR-11411). There are no restrictions on data availability and all materials are available from S Weyand (sw644@cam.ac.uk).

The following datasets were generated:

| Author(s) | Year | Dataset title | Dataset URL | Database and Identifier |
|---|---|---|---|---|
| Weyand SN, Matusevicius M, Yamashita K | 2024 | Molecular Mechanism of trypanosomal AQP2 | https://doi.org/10.2210/pdb8OFZ/pdb | Worldwide Protein Data Bank, 10.2210/pdb8OFZ/pdb |
| Weyand SN, Matusevicius M, Yamashita K | 2025 | Molecular Mechanism of trypanosomal AQP2 | https://www.ebi.ac.uk/emdb/EMD-16864 | EMDataBank, EMD-16864 |

*Continued on next page*

*Continued*

| Author(s) | Year | Dataset title | Dataset URL | Database and Identifier |
|---|---|---|---|---|
| Weyand SN, Gragera M | 2025 | Molecular mechanism of *Trypanosoma brucei* Aquaporin 2 with Glycerol | https://www.ebi.ac.uk/empiar/EMPIAR-11410 | EMPIAR, EMPIAR-11410 |
| Weyand SN, Matusevicius M, Yamashita K | 2024 | Molecular Mechanism of trypanosomal AQP2 | https://doi.org/10.2210/pdb8OFY/pdb | Worldwide Protein Data Bank, 10.2210/pdb8OFY/pdb |
| Weyand SN, Matusevicius M, Yamashita K | 2025 | Molecular Mechanism of trypanosomal AQP2 | https://www.ebi.ac.uk/emdb/EMD-16863 | EMDataBank, EMD-16863 |
| Weyand SN, Gragera M | 2025 | Molecular mechanism of *Trypanosoma brucei* Aquaporin 2 with pentamidine | https://www.ebi.ac.uk/empiar/EMPIAR-11412 | EMPIAR, EMPIAR-11412 |
| Weyand SN, Matusevicius M, Yamashita K | 2024 | Molecular Mechanism of trypanosomal AQP2 | https://doi.org/10.2210/pdb8OFX/pdb | Worldwide Protein Data Bank, 10.2210/pdb8OFX/pdb |
| Weyand SN, Matusevicius M, Yamashita K | 2025 | Molecular Mechanism of trypanosomal AQP2 | https://www.ebi.ac.uk/emdb/EMD-16862 | EMDataBank, EMD-16862 |
| Weyand SN, Gragera M | 2025 | Molecular mechanism of *Trypanosoma brucei* Aquaporin 2 with melarsoprol | https://www.ebi.ac.uk/empiar/EMPIAR-11411 | EMPIAR, EMPIAR-11411 |

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
