## [Editor Report · eLife Assessment]

In this **important** study, the authors set out to determine the molecular interactions between the AQP2 from *Trypanosoma brucei* (TbAQP2) and the trypanocidal drugs pentamidine and melarsoprol to understand how TbAQP2 mutations lead to drug resistance. Using cryo-EM, molecular dynamics simulations, and lysis assays the authors present **convincing** evidence that mutations in TbAQP2 make permeation of trypanocidal drugs energetically less favourable, and that this impacts the ability of drugs to achieve a therapeutic dose. Overall, this data will be of interest for those working on aquaporins, and development of trypanosomiasis drugs as well as drugs targeting aquaporins in general.

---

## [Referee Report · Reviewer #1 (Public review)]

This study presents cryoEM-derived structures of the Trypanosome aquaporin AQP2, in complex with its natural ligand, glycerol, as well as two trypanocidal drugs, pentamidine and melarsoprol, which use AQP2 as an uptake route. The structures are high quality and the density for the drug molecules is convincing, showing a binding site in the centre of the AQP2 pore.

The authors then continue to study this system using molecular dynamics simulations. Their simulations indicate that the drugs can pass through the pore and identify a weak binding site in the centre of the pore, which corresponds with that identified through cryoEM analysis. They also simulate the effect of drug resistance mutations which suggests that the mutations reduce the affinity for drugs and therefore might reduce the likelihood that the drugs enter into the centre of the pore, reducing the likelihood that they progress through into the cell.

While the cryoEM and MD studies are well conducted, it is a shame that the drug transport hypothesis was not tested experimentally. For example, did they do cryoEM with AQP2 with drug resistance mutations and see if they could see the drugs in these maps? They might not bind, but another possibility is that the binding site shifts, as seen in Chen et al? Do they have an assay for measuring drug binding? I think that some experimental validation of the drug binding hypothesis would strengthen this paper. The authors describe in their response why these experiments are challenging.

---

## [Referee Report · Reviewer #2 (Public review)]

Summary:

The authors present 3.2-3.7 Å cryo-EM structures of *Trypanosoma brucei* aquaglyceroporin-2 (TbAQP2) bound to glycerol, pentamidine or melarsoprol and combine them with extensive all-atom MD simulations to explain drug recognition and resistance mutations. The work provides a persuasive structural rationale for (i) why positively selected pore substitutions enable diamidine uptake, and (ii) how clinical resistance mutations weaken the high-affinity energy minimum that drives permeation. These insights are valuable for chemotherapeutic re-engineering of diamidines and aquaglyceroporin-mediated drug delivery.

My comments are on the MD part

Strengths:

The study

Integrates complementary cryo-EM, equilibrium and applied voltage MD simulations, and umbrella-sampling PMFs, yielding a coherent molecular-level picture of drug permeation.

Offers direct structural rationalisation of long-standing resistance mutations in trypanosomes, addressing an important medical problem.

Comments on revisions:

Most of the weaknesses have been resolved during the revision process.

---

## [Referee Report · Reviewer #3 (Public review)]

Summary:

Recent studies have established that trypanocidal drugs, including pentamidine and melarsoprol, enter the trypanosomes via the glyceroaquaporin AQP2 (TbAQP2). Interestingly, drug resistance in trypanosomes is, at least in part, caused by recombination with the neighbouring gene, AQP3, which is unable to permeate pentamidine or melarsoprol. The effect of the drugs on cells expressing chimeric proteins is significantly reduced. In addition, controversy exists regarding whether TbAQP2 permeates the drugs like an ion channel, or whether it serves as a receptor that triggers downstream processes upon drug binding. In this study the authors set out to achieve these objectives: (1) to understand the molecular interactions between TbAQP2 and glycerol, pentamidine, and melarsoprol, and (2) to determine the mechanism by which mutations that arise from recombination with TbAQP3 result in reduced drug permeation.

The cryo-EM structures provide details of glycerol and drug binding, and show that glycerol and the drugs occupy the same space within the pore. Finally, MD simulations and lysis assays are employed to determine how mutations in TbAQP2 result in reduced permeation of drugs by making entry and exit of the drug relatively more energy-expensive. Overall, the strength of evidence used to support the author's claims is solid.

Strengths:

The cryo-EM portion of the study is strong, and while the overall resolution of the structures is in the 3.5Å range, the local resolution within the core of the protein and the drug binding sites is considerably higher (~2.5Å).

I also appreciated the MD simulations on the TbAQP2 mutants and the mechanistic insights that resulted from this data.

Weaknesses:

(1) The authors do not provide any experimental validation the drug binding sites in TbAQP2 due to lacking resources. However, the claims have been softened in the revised paper.

---

## [Author Response]

The following is the authors’ response to the original reviews.

**Reviewer #1 (Public review):**
This study presents cryoEM-derived structures of the Trypanosome aquaporin AQP2, in complex with its natural ligand, glycerol, as well as two trypanocidal drugs, pentamidine and melarsoprol, which use AQP2 as an uptake route. The structures are high quality, and the density for the drug molecules is convincing, showing a binding site in the centre of the AQP2 pore.The authors then continue to study this system using molecular dynamics simulations. Their simulations indicate that the drugs can pass through the pore and identify a weak binding site in the centre of the pore, which corresponds with that identified through cryoEM analysis. They also simulate the effect of drug resistance mutations, which suggests that the mutations reduce the affinity for drugs and therefore might reduce the likelihood that the drugs enter into the centre of the pore, reducing the likelihood that they progress through into the cell.While the cryoEM and MD studies are well conducted, it is a shame that the drug transport hypothesis was not tested experimentally. For example, did they do cryoEM with AQP2 with drug resistance mutations and see if they could see the drugs in these maps? They might not bind, but another possibility is that the binding site shifts, as seen in Chen et al.

TbAQP2 from the drug-resistant mutants does not transport either melarsoprol or pentamidine and there was thus no evidence to suggest that the mutant TbAQP2 channels could bind either drug. Moreover, there is not a single mutation that is characteristic for drug resistance in TbAQP2: references 12–15 show a plethora of chimeric AQP2/3 constructs in addition to various point mutations in laboratory strains and field isolates. In reference 17 we describe a substantial number of SNPs that reduced pentamidine and melarsoprol efficacy to levels that would constitute clinical resistance to acceptable dosage regimen. It thus appears that there are many and diverse mutations that are able to modify the protein sufficiently to induce resistance, and likely in multiple different ways, including the narrowing of the pore, changes to interacting amino acids, access to the pore etc. We therefore did not attempt to determine the structures of the mutant channels because we did not think that in most cases we would see any density for the drugs in the channel, and we would be unable to define ‘the’ resistance mechanism if we did in the case of one individual mutant TbAQP2. Our MD data suggests that pentamidine binding affinity is in the range of 50-300 µM for the mutant TbAQP2s selected for that test (I110W and L258Y/L264R), i.e. >1000-fold higher than TbAQP2WT. Thus these structures will be exceedingly challenging to determine with pentamidine in the pore but, of course, until the experiment has been tried we will not know for sure.

Do they have an assay for measuring drug binding?

We tried many years ago to develop a ^3^H-pentamidine binding assay to purified wild type TbAQP2 but we never got satisfactory results even though the binding should be in the doubledigit nanomolar range. This may be for any number of technical reasons and could also be partly because flexible di-benzamidines bind non-specifically to proteins at µM concentrations giving rise to high background. Measuring binding to the mutants was not tested given that they would be binding pentamidine in the µM range. If we were to pursue this further, then isothermal titration calorimetry (ITC) may be one way forward as this can measure µM affinity binding using unlabelled compounds, although it uses a lot of protein and background binding would need to be carefully assessed; see for example our work on measuring tetracycline binding to the tetracycline antiporter TetAB (https://doi.org/10.1016/j.bbamem.2015.06.026). Membrane proteins are also particularly tricky for this technique as the chemical activity of the protein solution must be identical to the chemical activity of the substrate solution which titrates in the molecule binding to the protein; this can be exceedingly problematic if any free detergent remains in the purified membrane protein. Another possibility may be fluorescence polarisation spectroscopy, although this would require fluorescently labelling the drugs which would very likely affect their affinity for TbAQP2 and how they interact with the wild type and mutant proteins – see the detailed SAR analysis in Alghamdi et al. 2020 (ref. 17). As you will appreciate, it would take considerable time and effort to set up an assay for measuring drug binding to mutants and is beyond the current scope of the current work.

I think that some experimental validation of the drug binding hypothesis would strengthen this paper. Without this, I would recommend the authors to soften the statement of their hypothesis (i.e, lines 65-68) as this has not been experimentally validated.

We agree with the referee that direct binding of drugs to the mutants would be very nice to have, but we have neither the time nor resources to do this. We have therefore softened the statement on lines 65-68 to read ‘Drug-resistant TbAQP2 mutants are still predicted to bind pentamidine, but the much weaker binding in the centre of the channel observed in the MD simulations would be insufficient to compensate for the high energy processes of ingress and egress, hence impairing transport at pharmacologically relevant concentrations.’

**Reviewer #2 (Public review):**
Summary:The authors present 3.2-3.7 Å cryo-EM structures of *Trypanosoma brucei* aquaglyceroporin-2 (TbAQP2) bound to glycerol, pentamidine, or melarsoprol and combine them with extensive allatom MD simulations to explain drug recognition and resistance mutations. The work provides a persuasive structural rationale for (i) why positively selected pore substitutions enable diamidine uptake, and (ii) how clinical resistance mutations weaken the high-affinity energy minimum that drives permeation. These insights are valuable for chemotherapeutic re-engineering of diamidines and aquaglyceroporin-mediated drug delivery.My comments are on the MD part.Strengths:The study(1) Integrates complementary cryo-EM, equilibrium, applied voltage MD simulations, and umbrella-sampling PMFs, yielding a coherent molecular-level picture of drug permeation.(2) Offers direct structural rationalisation of long-standing resistance mutations in trypanosomes, addressing an important medical problem.Weaknesses:Unphysiological membrane potential. A field of 0.1 V nm ¹ (~1 V across the bilayer) was applied to accelerate translocation. From the traces (Figure 1c), it can be seen that the translocation occurred really quickly through the channel, suggesting that the field might have introduced some large changes in the protein. The authors state that they checked visually for this, but some additional analysis, especially of the residues next to the drug, would be welcome.

This is a good point from the referee, and we thank them for raising it. It is common to use membrane potentials in simulations that are higher than the physiological value, although these are typically lower than used here. The reason we used the higher value was to speed sampling and it still took 1,400 ns for transport in the physiologically correct direction, and even then, only in 1/3 repeats. Hence this choice of voltage was probably necessary to see the effect. The exceedingly slow rate of pentamidine permeation seen in the MD simulation was consistent with the experimental observations, as discussed in Alghamdi et al (2020) [ref. 17] where we estimated that TbAQP2-mediated pentamidine uptake in *T. brucei* bloodstream forms proceeds at just 9.5×10^5^ molecules/cell/h; the number of functional TbAQP2 units in the plasma membrane is not known but their location is limited to the small flagellar pocket (Quintana et al. PLoS Negl Trop Dis 14, e0008458 (2020)).

The referee is correct that it is important to make sure that the applied voltage is not causing issues for the protein, especially for residues in contact with the drug. We have carried out RMSF analysis to better test this. The data show that comparing our simulations with the voltage applied to the monomeric MD simulations + PNTM with no voltage reveals little difference in the dynamics of the drug-contacting residues.

We have added these new data as Supplementary Fig12b with a new legend (lines1134-1138)

‘b, RMSF calculations were run on monomeric TbAQP2 with either no membrane voltage or a 0.1V nm^-1^ voltage applied (in the physiological direction). Shown are residues in contact with the pentamidine molecule, coloured by RMSF value. RMSF values are shown for residues Leu122, Phe226, Ile241, and Leu264. The data suggest the voltage has little impact on the flexibility or stability of the pore lining residues.’

We have also added the following text to the manuscript (lines 524-530):

‘Membrane potential simulations were run using the computational electrophysiology protocol. An electric field of 0.1 V/nm was applied in the z-axis dimension only, to create a membrane potential of about 1 V (see Fig. S10a). Note that this is higher than the physiological value of 87.1 ± 2.1 mV at pH 7.3 in bloodstream *T. brucei*, and was chosen to improve the sampling efficiency of the simulations. The protein and lipid molecules were visually confirmed to be unaffected by this voltage, which we quantify using RMSF analysis on pentamidine-contacting residues (Fig. S12b).’

Based on applied voltage simulations, the authors argue that the membrane potential would help get the drug into the cell, and that a high value of the potential was applied merely to speed up the simulation. At the same time, the barrier for translocation from PMF calculations is ~40 kJ/mol for WT. Is the physiological membrane voltage enough to overcome this barrier in a realistic time? In this context, I do not see how much value the applied voltage simulations have, as one can estimate the work needed to translocate the substrate on PMF profiles alone. The authors might want to tone down their conclusions about the role of membrane voltage in the drug translocation.

We agree that the PMF barriers are considerable, however we highlight that other studies have seen similar landscapes, e.g. PMID 38734677 which saw a barrier of ca. 10-15 kcal/mol (ca. 4060 kJ/mol) for PNTM transversing the channel. This was reduced by ca. 4 kcal/mol when a 0.4 V nm ¹ membrane potential was applied, so we expect a similar effect to be seen here.

We have updated the Results to more clearly highlight this point and added the following text (lines 274-275):

We note that previous studies using these approaches saw energy barriers of a similar size, and that these are reduced in the presence of a membrane voltage[17,31].’

Pentamidine charge state and protonation. The ligand was modeled as +2, yet pKa values might change with the micro-environment. Some justification of this choice would be welcome.

Pentamidine contains two diamidine groups and each are expected to have a pKa above 10 in solution (PMID: 20368397), suggesting that the molecule will carry a +2 charge. Using the +2 charge is also in line with previous MD studies (PMID: 32762841). We have added the following text to the Methods (lines 506-509):

‘The pentamidine molecule used existing parameters available in the CHARMM36 database under the name PNTM with a charge state of +2 to reflect the predicted pKas of >10 for these groups [73] and in line with previous MD studies[17].’

We note that accounting for the impact of the microenvironment is an excellent point – future studies might employ constant pH calculations to address this.

The authors state that this RMSD is small for the substrate and show plots in Figure S7a, with the bottom plot being presumably done for the substrate (the legends are misleading, though), levelling off at ~0.15 nm RMSD. However, in Figure S7a, we see one trace (light blue) deviating from the initial position by more than 0.2 nm - that would surely result in an RMSD larger than 0.15, but this is somewhat not reflected in the RMSD plots.

The bottom plot of Fig. S9a (previously Fig. S7a) is indeed the RMSD of the drug (in relation to the protein). We have clarified the legend with the following text (lines 1037-1038): ‘… or for the pentamidine molecule itself, i.e. in relation to the Cα of the channel (bottom).’

With regards the second comment, we assume the referee is referring to the light blue trace from Fig S9c. These data are actually for the monomeric channel rather than the tetramer. We apologise for not making this clearer in the legend. We have added the word ‘monomeric’ (line 1041).

**Reviewer #3 (Public review):**
Summary:Recent studies have established that trypanocidal drugs, including pentamidine and melarsoprol, enter the trypanosomes via the glyceroaquaporin AQP2 (TbAQP2). Interestingly, drug resistance in trypanosomes is, at least in part, caused by recombination with the neighbouring gene, AQP3, which is unable to permeate pentamidine or melarsoprol. The effect of the drugs on cells expressing chimeric proteins is significantly reduced. In addition, controversy exists regarding whether TbAQP2 permeates drugs like an ion channel, or whether it serves as a receptor that triggers downstream processes upon drug binding. In this study the authors set out to achieve three objectives:(1) to determine if TbAQP2 acts as a channel or a receptor,

We should clarify here that this was not an objective of the current manuscript as the transport activity has already been extensively characterised in the literature, as described in the introduction.

(2) to understand the molecular interactions between TbAQP2 and glycerol, pentamidine, and melarsoprol, and(3) to determine the mechanism by which mutations that arise from recombination with TbAQP3 result in reduced drug permeation.Indeed, all three objectives are achieved in this paper. Using MD simulations and cryo-EM, the authors determine that TbAQP2 likely permeates drugs like an ion channel. The cryo-EM structures provide details of glycerol and drug binding, and show that glycerol and the drugs occupy the same space within the pore. Finally, MD simulations and lysis assays are employed to determine how mutations in TbAQP2 result in reduced permeation of drugs by making entry and exit of the drug relatively more energy-expensive. Overall, the strength of evidence used to support the author's claims is solid.Strengths:The cryo-EM portion of the study is strong, and while the overall resolution of the structures is in the 3.5Å range, the local resolution within the core of the protein and the drug binding sites is considerably higher (~2.5Å).I also appreciated the MD simulations on the TbAQP2 mutants and the mechanistic insights that resulted from this data.Weaknesses:(1) The authors do not provide any empirical validation of the drug binding sites in TbAQP2. While the discussion mentions that the binding site should not be thought of as a classical fixed site, the MD simulations show that there's an energetically preferred slot (i.e., high occupancy interactions) within the pore for the drugs. For example, mutagenesis and a lysis assay could provide us with some idea of the contribution/importance of the various residues identified in the structures to drug permeation. This data would also likely be very valuable in learning about selectivity for drugs in different AQP proteins.

On a philosophical level, we disagree with the requirement for ‘validation’ of a structure by mutagenesis. It is unclear what such mutagenesis would tell us beyond what was already shown experimentally through ^3^H-pentamidine transport, drug sensitivity and lysis assays i.e. a given mutation will impact permeation to a certain extent. But on the structural level, what does mutagenesis tell us? If a bulky aromatic residue that makes many van der Waals interactions with the substrate is changed to an alanine residue and transport is reduced, what does this mean? It would confirm that the phenylalanine residue is very likely indeed making van der Waals contacts to the substrate, but we knew that already from the WT structure. And if it doesn’t have any effect? Well, it could mean that the van der Waals interactions with that particular residue are not that important or it could be that the substrate has changed its positions slightly in the channel and the new pose has similar energy of interactions to that observed in the wild type channel. Regardless of the result, any data from mutagenesis would be open to interpretation and therefore would not impact on the conclusions drawn in this manuscript. We might not learn anything new unless all residues interacting with the substrate are mutated, the structure of each mutant was determined and MD simulations were performed for all, which is beyond the scope of this work. Even then, the value for understanding clinical drug resistance would be limited, as this phenomenon has been linked to various chimeric rearrangements with adjacent TbAQP3 (references 12–15), each with a structure distinct from TbAQP2 with a single SNP. We also note that the recent paper by Chen et al. did not include any mutagenesis of the drug binding sites in TbAQP2 in their analysis of TbAQP2, presumably for similar reasons as discussed above.

(2) Given the importance of AQP3 in the shaping of AQP2-mediated drug resistance, I think a figure showing a comparison between the two protein structures/AlphaFold structures would be beneficial and appropriate

We agree that the comparison is of considerably interest and would contribute further to our understanding of the unique permeation capacities of TbAQP2. As such, we followed the reviewer’s suggestion and made an AlphaFold model of TbAQP3 and compared it to our structures of TbAQP2. The RMSD is 0.6 Å to the pentamidine-bound TbAQP2, suggesting that the fold of TbAQP3 has been predicted well, although the side chain rotamers cannot be assessed for their accuracy. Previous work has defined the selectivity filter of TbAQP3 to be formed by W102, R256, Y250. The superposition of the TbAQP3 model and the TbAQP2 pentamidine-bound structure shows that one of the amine groups is level with R256 and that there is a clash with Y250 and the backbone carbonyl of Y250, which deviates in position from the backbone of TbAQP2 in this region. There is also a clash with Ile252.

Although these observations are indeed interesting, on their own they are highly preliminary and extensive further work would be necessary to draw any convincing conclusions regarding these residues in preventing uptake of pentamidine and melarsoprol. The TbAQP3 AlphaFold model would need to be verified by MD simulations and then we would want to look at how pentamidine would interact with the channel under different experimental conditions like we have done with TbAQP2. We would then want to mutate to Ala each of the residues singly and in combination and assess them in uptake assays to verify data from the MD simulations. This is a whole new study and, given the uncertainties surrounding the observations of just superimposing TbAQP2 structure and the TbAQP3 model, we feel that, regrettably, this is just too speculative to add to our manuscript.

(3) A few additional figures showing cryo-EM density, from both full maps and half maps, would help validate the data.

Two new Supplementary Figures have been made, on showing the densities for each of the secondary structure elements (the new Figure S5) and one for the half maps showing the ligands (the new Figure S6). All the remaining supplementary figures have been renamed accordingly.

(4) Finally, this paper might benefit from including more comparisons with and analysis of data published in Chen et al (doi.org/10.1038/s41467-024-48445-4), which focus on similar objectives. Looking at all the data in aggregate might reveal insights that are not obvious from either paper on their own. For example, melarsoprol binds differently in structures reported in the two respective papers, and this may tell us something about the energy of drug-protein interactions within the pore.

We already made the comparisons that we felt were most pertinent and included a figure (Fig. 5) to show the difference in orientation of melarsoprol in the two structures. We do not feel that any additional comparison is sufficiently interesting to be included. As we point out, the structures are virtually identical (RMSD 0.6 Å) and therefore there are no further mechanistic insights we would like to make beyond the thorough discussion in the Chen et al paper.

**Reviewer #1 (Recommendations for the authors):**
(1) Line 65 - I don't think that the authors have tested binding experimentally, and so rather than 'still bind', I think that 'are still predicted to bind' is more appropriate.

Changed as suggested

(2) Line 69 - remove 'and'

Changed as suggested

(3) Line 111 - clarify that it is the protein chain which is 'identical'. Ligands not.

Changed to read ‘The cryo-EM structures of TbAQP2 (excluding the drugs/substrates) were virtually identical…

(4) Line 186 - make the heading of this section more descriptive of the conclusion than the technique?

We have changed the heading to read: ‘Molecular dynamics simulations show impaired pentamidine transport in mutants’

**Reviewer #2 (Recommendations for the authors):**
(1) Methods - a rate of 1 nm per ns is mentioned for pulling simulations, is that right?

Yes, for the generation of the initial frames for the umbrella sampling a pull rate of 1 nm/ns was used in either an upwards or downwards z-dimension

(2) Figure S9 and S10 have their captions swapped.

The captions have been swapped to their proper positions.

(3) Methods state "40 ns per window" yet also that "the first 50 ns of each window was discarded as equilibration".

Well spotted - this line should have read “the first 5 ns of each window was discarded as equilibration”. This has been corrected (line 541).

**Reviewer #3 (Recommendations for the authors):**
(1) Abstract, line 68-70: incomplete sentence.

The sentence has been re-written: ‘The structures of drug-bound TbAQP2 represent a novel paradigm for drug-transporter interactions and are a new mechanism for targeting drugs in pathogens and human cells.

(2) Line 312-313: The paper you mention here came out in May 2024 - a year ago. I appreciate that they reported similar structural data, but for the benefit of the readers and the field, I would recommend a more thorough account of the points by which the two pieces of work differ. Is there some knowledge that can be gleaned by looking at all the data in the two papers together? For example, you report a glycerol-bound structure while the other group provides an apo one. Are there any mechanistic insights that can be gained from a comparison?

We already made the comparisons that we felt were most pertinent and included a figure (Fig. 5) to show the difference in orientation of melarsoprol in the two structures. We do not feel that any additional comparison is sufficiently interesting to be included. As we point out, the structures are virtually identical (RMSD 0.6 Å) and therefore there are no further mechanistic insights we would like to make beyond the thorough discussion in the Chen et al paper.

(3) Similarly, you can highlight the findings from your MD simulations on the TbAQP2 drug resistance mutants, which are unique to your study. How can this data help with solving the drug resistance problem?

New drugs will need to be developed that can be transported by the mutant chimera AQP2s and the models from the MD simulations will provide a starting point for molecular docking studies. Further work will then be required in transport assays to optimise transport rather than merely binding. However, the fact that drug resistance can also arise through deletion of the AQP2 gene highlights the need for developing new drugs that target other proteins.

(4) A glaring question that one has as a reader is why you have not attempted to solve the structures of the drug resistance mutants, either in complex with the two compounds or in their apo/glycerol-bound form? To be clear, I am not requesting this data, but it might be a good idea to bring this up in the discussion.

TbAQP2 containing the drug-resistant mutants does not transport either melarsoprol or pentamidine (Munday et al., 2014; Alghamdi et al., 2020); there was thus no evidence to suggest that the mutant TbAQP2 channels could bind either drug. We therefore did not attempt to determine the structures of the mutant channels because we did not think that we would see any density for the drugs in the channel. Our MD data suggests that pentamidine binding affinity is in the range of 50-300 µM for the mutant TbAQP2, supporting the view that getting these structures would be highly challenging, but of course until the experiment is tried we will not know for sure.

We also do not think we would learn anything new about doing structures of the drug-free structures of the transport-negative mutants of TbAQP2. The MD simulations have given novel insights into why the drugs are not transported and we would rather expand effort in this direction and look at other mutants rather than expend further effort in determining new structures.

(5) Line 152-156: Is there a molecular explanation for why the TbAQP2 has 2 glycerol molecules captured in the selectivity filter while the PfAQP2 and the human AQP7 and AQP10 have 3?

The presence of glycerol molecules represents local energy minima for binding, which will depend on the local disposition of appropriate hydrogen bonding atoms and hydrophobic regions, in conjunction with the narrowness of the channel to effectively bind glycerol from all sides. It is noticeable that the extracellular region of the channel is wider in TbAQP2 than in AQP7 and AQP10, so this may be one reason why additional ordered glycerol molecules are absent, and only two are observed. Note also that the other structures were determined by X-ray crystallography, and the environment of the crystal lattice may have significantly decreased the rate of diffusion of glycerol, increasing the likelihood of observing their electron densities.

(6) I would also think about including the 8JY7 (TbAQP2 apo) structure in your analysis.

We included 8JY7 in our original analyses, but the results were identical to 8JY6 and 8JY8 in terms of the protein structure, and, in the absence of any modelled substrates in 8JY7 (the interesting part for our manuscript), we therefore have not included the comparison.

(7) I also think, given the importance of AQP3 in this context, it would be really useful to have a comparison with the AQP3 AlphaFold structure in order to examine why it does not permeate drugs.

We made an AlphaFold model of TbAQP3 and compared it to our structures of TbAQP2. The RMSD is 0.6 Å to the pentamidine-bound TbAQP2, suggesting that the fold of TbAQP3 has been predicted well, although the side chain rotamers cannot be assessed for their accuracy. Previous work has defined the selectivity filter of TbAQP3 to be formed by W102, R256, Y250. The superposition of the TbAQP3 model and the TbAQP2 pentamidine-bound structure shows that one of the amine groups is level with R256 and that there is a clash with Y250 and the backbone carbonyl of Y250, which deviates in position from the backbone of TbAQP2 in this region. There is also a clash with Ile252.

Although these observations are interesting, on their own they are preliminary in the extreme and extensive further work will be necessary to draw any convincing conclusions regarding these residues in preventing uptake of pentamidine and melarsoprol. The TbAQP3 AlphaFold model would need to be verified by MD simulations and then we would want to look at how pentamidine would interact with the channel under different experimental conditions like we have done with TbAQP2. We would then want to mutate to Ala each of the residues singly and in combination and assess them in uptake assays to verify data from the MD simulations. This is a whole new study and, given the uncertainties surrounding the observations of just superimposing TbAQP2 structure and the TbAQP3 model, we feel this is just too speculative to add to our manuscript.

(8) To validate the densities representing glycerol and the compounds, you should show halfmap densities for these.

A new figure, Fig S6 has been made to show the half-map densities for the glycerol and drugs.

(9) I would also like to see the density coverage of the individual helices/structural elements.

A new figure, Fig S5 has been made to show the densities for the structural elements.

(10) While the LigPlot figure is nice, I think showing the data (including the cryo-EM density) is necessary validation.

The LigPlot figure is a diagram (an interpretation of data) and does not need the densities as these have already been shown in Fig. 1c (the data).

(11) I would recommend including a figure that illustrates the points described in lines 123-134.

All of the points raised in this section are already shown in Fig. 2a, which was referred to twice in this section. We have added another reference to Fig.2a on lines 134-135 for completeness.

(12) Line 202: I would suggest using "membrane potential/voltage" to avoid confusion with mitochondrial membrane potential.

We have changed this to ‘plasma membrane potential’ to differentiate it from mitochondrial membrane potential.

(13) Figure 4: Label C.O.M. in the panels so that the figure corresponds to the legend.

We have altered the figure and added and explanation in the figure legend (lines 716-717):

‘Cyan mesh shows the density of the molecule across the MD simulation. and the asterisk shows the position of the centre of mass (COM).’

(14) Figure S2: Panels d and e appear too similar, and it is difficult to see the stick representation of the compound. I would recommend either using different colours or showing a close-up of the site.

We have clarified the figure by including two close-up views of the hot-spot region, one with melarsoprol overlaid and one with pentamidine overlaid

(15) Figure S2: Typo in legend: 8YJ7 should be 8JY7.

Changed as suggested

(16) Figure S3 and Figure S4: Please clarify which parts of the process were performed in cryoSPARC and which in Relion.

Figure S3 gives an overview of the processing and has been simplified to give the overall picture of the procedures. All of the details were included in the Methods section as other programmes are used, not just cryoSPARC and Relion. Given the complexities of the processing, we have referred the readers to the Methods section rather than giving confusing information in Fig. S3.

We have updated the figure legend to Fig. S4 as requested.

(17) Figure S9 and Figure S10: The legends are swapped in these two figures.

The captions have been swapped to their proper positions.

(18) For ease of orientation and viewing, I would recommend showing a vertical HOLE plot aligned with an image of the AQP2 pore.

The HOLE plot has been re-drawn as suggest (Fig. S2)